# Tight Rates for Bandit Control Beyond Quadratics

**Y. Jennifer Sun**
Princeton University
ys7849@princeton.edu

**Zhou Lu**
Princeton University
zhoul@princeton.edu

## Abstract

Unlike classical control theory, such as Linear Quadratic Control (LQC), real-world control problems are highly complex. These problems often involve adversarial perturbations, bandit feedback models, and non-quadratic, adversarially chosen cost functions. A fundamental yet unresolved question is whether optimal regret can be achieved for these general control problems. The standard approach to addressing this problem involves a reduction to bandit convex optimization with memory. In the bandit setting, constructing a gradient estimator with low variance is challenging due to the memory structure and non-quadratic loss functions.

In this paper, we provide an affirmative answer to this question. Our main contribution is an algorithm that achieves an $\tilde{O}(\sqrt{T})$ optimal regret for bandit non-stochastic control with strongly-convex and smooth cost functions in the presence of adversarial perturbations, improving the previously known $\tilde{O}(T^{2/3})$ regret bound from (Cassel and Koren, 2020). Our algorithm overcomes the memory issue by reducing the problem to Bandit Convex Optimization (BCO) without memory and addresses general strongly-convex costs using recent advancements in BCO from (Suggala et al., 2024). Along the way, we develop an improved algorithm for BCO with memory, which may be of independent interest.

## 1 Introduction

Optimal control lies at the heart of engineering and operations research, with applications ranging from launching spacecraft to stabilizing economies. The theory of optimal control is a well-established field with a rich history, dating back to 1868 when James Clerk Maxwell analyzed governors, and flourishing in the mid-20th century with the development of dynamic programming by (Bellman, 1954) and the Kalman filter by (Kalman, 1960).

Classic optimal control theory studies the problem where a controller interacts with an environment, according to a (partially observable) linear time-invariant (LTI) dynamical system:

$$x_{t+1} = Ax_t + Bu_t + w_t, \quad y_t = Cx_t + e_t, \tag{1}$$

where $A, B, C$ are the dynamics governing the system, $x_t, u_t, y_t, w_t, e_t$ represent the system state, control input, observation, perturbation, and observation noise at time $t$, respectively. At each time $t$, the controller observes $y_t$ and chooses a control $u_t$, incurring a cost $c_t(y_t, u_t)$ based on the current observation and control. The system then evolves according to Eq. (1) to reach the next state $x_{t+1}$.

The theory of optimal control (e.g., LQC) typically relies on three key assumptions on the setting: the perturbation $w_t$ is stochastic, the cost function $c_t$ is quadratic and known in advance, and the function $c_t$ is observable to the controller. The linear-quadratic regulator (LQR) (Kalman et al., 1960) provides a closed-form optimal solution under these conditions, representing a pinnacle of classical control theory.

However, these assumptions are often too idealistic for practical scenarios. In real-world control problems, the perturbations can be adversarial, the feedback model can be bandit, and the cost

38th Conference on Neural Information Processing Systems (NeurIPS 2024).

function can be non-quadratic. This is evident in applications such as autonomous vehicle navigation, advertisement placement, and traffic signal control. This discrepancy between theory and practice raises a fundamental question for developing a more general control theory:

*Can we devise algorithms with provable guarantees for LTI control problems with adversarial perturbations, bandit feedback models, and non-quadratic cost?*

Recent research in online non-stochastic control (see (Hazan and Singh, 2022) for a survey) aims to address this broader goal by relaxing two of the standard assumptions: (1) the cost $c_t$ can be time-varying, non-quadratic convex functions unknown to the controller; (2) the perturbation $w_t$ and the observation noise $e_t$ can be adversarially chosen.

The natural performance metric in this context is *regret*, defined as the excess cost incurred by the controller compared to the best control policy in a benchmark policy class $\Pi$:

$$\text{Regret}_T^\Pi(\texttt{controller}) = \sum_{t=1}^T c_t(y_t, u_t) - \min_{\pi \in \Pi} \sum_{t=1}^T c_t(y_t^\pi, u_t^\pi), \tag{2}$$

where $(y_t, u_t)$ is the observation-control pair reached by executing the controller at time $t$, and $(y_t^\pi, u_t^\pi)$ is the observation-control pair under the policy $\pi$ at time $t$. An optimal $\tilde{O}(\sqrt{T})$ regret was obtained by (Agarwal et al., 2019a) with the Gradient Perturbation Controller (GPC) algorithm.

Several works in online control have made further progress towards the general question. (Cassel and Koren, 2020; Sun et al., 2024) obtained optimal regret under bandit feedback for strongly convex and smooth cost when the perturbation $w_t$ and observation noise $e_t$ are semi-adversarial (i.e. they contain an additive stochastic component that admits covariance matrices with least singular value bounded from below). (Cassel and Koren, 2020) also showed a sub-optimal $\tilde{O}(T^{2/3})$ regret bound for fully adversarial perturbations. More recently, the advancement of (Suggala et al., 2024) showed for the first time that an optimal regret of $\tilde{O}(\sqrt{T})$ is achievable for strongly convex and smooth **quadratic** costs in the presence of adversarial perturbations and observation noises.

Still, no previous work has simultaneously addressed all three challenges with an optimal regret guarantee, which was left as an open problem by (Suggala et al., 2024). The main challenge lies in how to construct a low-variance gradient estimator under bandit feedback, with the memory structure and non-quadratic cost. Due to the $\Omega(T^{2/3})$ regret lower bound for general BCO with memory by (Suggala et al., 2024), exploiting the special affine memory structure in control problems is crucial to achieving optimal regret for general convex cost.

In this work, we provide the first affirmative answer to this general question by devising an algorithm that handles all three challenges with an optimal $\tilde{O}(\sqrt{T})$ regret bound. Our approach involves reducing the problem to no-memory BCO, which circumvents the high-dimensional estimator issue for non-quadratic costs. We then leverage a special curvature structure of the loss function induced by general strongly convex and smooth costs to obtain the optimal regret guarantee. Our result serves as a preliminary step toward fully solving the general control problem.

## 1.1 Technical Overview

Several previous works have addressed the bandit non-stochastic control problem, but none achieved optimal regret across all three generalities due to the following technical challenges:

1. **The necessity of curvature**: The first work tackling the bandit non-stochastic control problem was (Cassel and Koren, 2020), which achieved an $\tilde{O}(T^{2/3})$ regret bound for smooth convex cost. However, such sub-optimality is arguably inevitable due to the $\tilde{\Omega}(T^{2/3})$ regret lower bound for BCO-M with smooth convex [1] cost (Suggala et al., 2024): all existing results on bandit non-stochastic control relies on reduction to BCO-M! Indeed, the algorithm of (Cassel and Koren, 2020) is based on online gradient descent (OGD) and ignores the geometry of cost. This is the reason why we introduce the assumption on strong convexity.

2. **Strong convexity is not enough**: Does strong convexity alone become an easy remedy to the control problem? Unfortunately, even if we assume the cost functions $c_t$ in the control

---

[1]In fact, the lower bound proved in (Suggala et al., 2024) even holds for quadratic loss functions.

| | Regret | Perturbation | Feedback | Loss type |
|---|---|---|---|---|
| (Agarwal et al., 2019a) | $\tilde{O}(\sqrt{T})$ | adversarial | full | strongly-convex |
| (Agarwal et al., 2019b) | $O(\log^7 T)$ | stochastic | full | convex |
| (Cassel and Koren, 2020) | $\tilde{O}(T^{2/3})$ | adversarial | bandit | convex smooth |
| (Cassel and Koren, 2020) | $\tilde{O}(\sqrt{T})$ | stochastic | bandit | strongly-convex smooth |
| (Sun et al., 2024) | $\tilde{O}(\sqrt{T})$ | semi-adv | bandit | str-conv smooth quadratic |
| (Suggala et al., 2024) | $\tilde{O}(\sqrt{T})$ | adversarial | bandit | str-conv smooth quadratic |
| This work | $\tilde{O}(\sqrt{T})$ | adversarial | bandit | strongly-convex smooth |

Table 1: Comparison of results. This work is the first to address all three generalities with an $\tilde{O}(\sqrt{T})$ optimal regret: (Agarwal et al., 2019a) addressed perturbation + loss, (Cassel and Koren, 2020) addressed feedback + loss, (Suggala et al., 2024) addressed perturbation + feedback. (Cassel and Koren, 2020) also obtained a sub-optimal $\tilde{O}(T^{2/3})$ regret for perturbation + feedback + loss.

      problem are strongly-convex, the induced loss functions in the BCO-M problem are not necessarily strongly-convex! It was observed by (Suggala et al., 2024) that the induced loss functions satisfy a property called $\kappa$-convexity, allowing for low-variance estimation of Hessian matrices, which is a key ingredient in Newton-based second-order update to exploit the curvature.

3. **Going beyond quadratic**: However, finding a low-biased first-order estimation of gradients remains a challenge for BCO-M, because the nice property on the unit sphere domain $\mathbb{S}_{d-1} := \{x \in \mathbb{R}^d \mid \|x\|_2 = 1\}$ of the exploration term is not preserved under Cartesian product: $\mathbb{S}_{d-1} \times \mathbb{S}_{d-1} \neq \mathbb{S}_{d \times d - 1}$. To handle this issue, (Suggala et al., 2024) relies on the quadratic cost assumption which admits an unbiased estimator of the divergence of the induced loss function. It's unclear how to handle general $\kappa$-convex smooth cost in BCO-M.

Similar to its full-information counterpart, we reduce the bandit non-stochastic control problem to bandit convex optimization with memory (BCO-M). To overcome these challenges, we further reduce the BCO-M problem to a no-memory BCO problem following (Cassel and Koren, 2020), to avoid the issue on the Cartesian product of $\mathbb{S}_{d-1}$. This allows for the use of a Newton-based BCO-M algorithm similar to (Suggala et al., 2024), which can handle general $\kappa$-convex smooth cost because the exploration domain is now $\mathbb{S}_{d-1}$. Besides this new approach, our method also includes novel algorithmic components and analysis.

## 1.2 Related Work

Optimal control theory (Bellman, 1954) concerns finding a control for a dynamical system such that some objective function is optimized. The basic form of optimal control is linear quadratic control (Kalman, 1960), in which a quadratic function is minimized in a linear first-order dynamical system with stochastic noise, whose solution is given by LQR.

Online non-stochastic control generalizes classic optimal control by considering adversarial perturbation and cost, with the metric of regret to compete with the best fixed policy in some benchmark class. The first work (Agarwal et al., 2019a) obtained an optimal regret bound for general convex cost by reduction to online convex optimization with memory (Anava et al., 2015), under stability assumptions on the system. For the line of works on the full information feedback setting, see (Hazan and Singh, 2022) for a survey.

Online non-stochastic control with bandit feedback were first considered by (Cassel and Koren, 2020; Gradu et al., 2020). Under a smoothness assumption on cost functions, the two works showed an $\tilde{O}(T^{2/3})$ and $\tilde{O}(T^{3/4})$ regret bound respectively. When the cost functions are additionally strongly-convex quadratics, (Sun et al., 2024) obtained an $\tilde{O}(\sqrt{T})$ regret, under semi-adversarial perturbations. (Yan et al., 2024) showed sublinear regret for cost functions with heterogeneous curvatures under the

same semi-adversarial perturbations. This restriction on perturbation was later removed by (Suggala et al., 2024) which achieved the same regret for adversarial perturbations.

Recent advancements on BCO (Lattimore and György, 2023; Suggala et al., 2021; Lattimore, 2024) considered Newton step updates to achieve improved regret bounds. In particular, (Suggala et al., 2024) also showed an $\tilde{\Omega}(T^{2/3})$ lower bound for bandit quadratic optimization with memory (BQO-M) without strong convexity. When making use of the affine memory structure, (Suggala et al., 2024) obtained an $\tilde{O}(\sqrt{T})$ regret bound for BQO-M.

## 2 Preliminary

### 2.1 Online non-stochastic control with bandit feedback

Consider the linear dynamical system in Eq. (1). At time $t \in \mathbb{N}$, the learner receives observation $y_t$ and outputs control $u_t$, to which the learner receives a scalar instantaneous cost $c_t(y_t, u_t)$ depending on both the current observation and the control, and no other information about $c_t$ is available. The perturbation $w_t$ and observation noise $e_t$ can be adversarially chosen. The learner's goal is to minimize regret over a fixed time horizon $T \in \mathbb{N}$ defined in Eq. (2).

Throughout this paper, all cost functions we consider are assumed to be twice continuously differentiable. Consistent with past literature (Sun et al., 2024; Suggala et al., 2024), we make the following assumptions on the cost functions.

**Assumption 1** (Cost curvature and regularity). *Let $d_y, d_u \in \mathbb{N}$ denote the dimensions of observation and control, respectively. The control cost function $c_t : \mathbb{R}^{d_y} \times \mathbb{R}^{d_u} \to \mathbb{R}_+$ satisfies that*

1. *(Curvature) $c_t$ is $\alpha_c$-strongly convex and $\beta_c$-smooth over some compact set $\mathcal{Y} \times \mathcal{U} \subset \mathbb{R}^{d_y} \times \mathbb{R}^{d_u}$ for some $\alpha_c, \beta_c > 0$, i.e., $\nabla c_t(x_2)^\top (x_1 - x_2) + \frac{\beta_c}{2} \|x_1 - x_2\|_2^2 \geq c_t(x_1) - c_t(x_2) \geq \nabla c_t(x_2)^\top (x_1 - x_2) + \frac{\alpha_c}{2} \|x_1 - x_2\|_2^2$ for any $x_1, x_2 \in \mathcal{Y} \times \mathcal{U}$.*

2. *(Gradient bounds) There exists some $G_c > 0$ such that the gradients satisfy $\|\nabla c_t(y, u)\|_2 \leq G_c \|(y, u)\|_2$ for any $y \in \mathcal{Y}, u \in \mathcal{U}$.*

We also make stability assumption on the LDS in Eq. (1) as well as norm bound on the perturbation and noise. These assumptions are standard in literature (e.g. (Hazan and Singh, 2022)) and necessary for deriving meaningful regret guarantees.

**Assumption 2** (Strong stabilizability and bounded dynamics). *$(A, B, C)$ that governs the LDS in Eq. (1) satisfies that $\max\{\|A\|_{\mathrm{op}}, \|B\|_{\mathrm{op}}, \|C\|_{\mathrm{op}}\} \leq \kappa_{sys}$ for some positive constant $\kappa_{sys}$. There exists $K \in \mathbb{R}^{d_u \times d_y}$ s.t. $A + BKC = HL^{-1}H$ for some $H \succ 0$ and $\max\{\|K\|_2, \|H\|_2, \|H^{-1}\|_2\} \leq \kappa$, $\|L\|_2 \leq 1 - \gamma$ for some $\kappa > 0$, $0 < \gamma \leq 1$. Such $K$ is called a $(\kappa, \gamma)$-stabilizing linear controller for the LDS.*

**Assumption 3** (Bounded perturbation and noise). *The perturbation and observation noise sequences satisfy the norm bound $\max_{t \in [T]} \{\max\{\|w_t\|_2, \|e_t\|_2\}\} \leq R_{w,e}$ for some $R_{w,e} > 0$.*

We assume the adversary chooses the cost functions and noise sequences in an oblivious way, i.e. they are chosen independently of the learner's decisions.

**Assumption 4** (Oblivious adversary). *The sequence of cost functions $\{c_t\}_{t=1}^T$ and the noise sequences $\{w_t\}_{t=1}^T, \{e_t\}_{t=1}^T$ are chosen by an oblivious adversary and does not depend on the control played by the learner.*

For partially observable systems, the standard comparator class in literature (Simchowitz et al., 2020) is the class of Disturbance Response Controllers (DRC). Essentially, this class considers controllers that choose linear combinations of past signals and observations. The signals are simply the would-be observations had a stabilizing controller $K$ been used from the beginning of the time. Formally, the class of DRC is given by the following definition:

**Definition 1** (DRC). (1) A disturbance response controller (DRC) is a policy $\pi_M$ parametrized by $m \in \mathbb{N}$ matrices $M = M^{[0:m-1]}$ in $\mathbb{R}^{d_u \times d_y}$ such that the control at time $t$ according to $\pi_M$ is

$$u_t(\pi_M) = K y_t + \sum_{j=0}^{m-1} M^{[j]} y_{t-j}(K),$$

where $K$ is a $(\kappa, \gamma)$-stabilizing linear controller, and $y_t^K$ is the would-be observation had the linear policy $K$ been carried out from the beginning of the time.

(2) A DRC policy class $\mathcal{M}(m, R_{\mathcal{M}})$, parameterized by $m \in \mathbb{N}, R_{\mathcal{M}} > 0$, is the set of all DRC controller of length $m$ and obeys the norm bound $\|M\|_{\ell_1, \mathrm{op}} := \sum_{j=0}^{m-1} \|M^{[j]}\|_{\mathrm{op}} \leq R_{\mathcal{M}}$.

A DRC policy always produces controls that ensure the system remains state bounded, the DRC class is expressive enough to approximate the class of linear policies of past observations (see Theorem 1 in (Simchowitz et al., 2020)).

## 2.2 Bandit convex optimization with memory (BCO-M)

We start with the general protocol of bandit convex optimization with memory (BCO-M). In the (improper) BCO-M problem, at each time $t$, the learner is asked to output a decision $z_t \in \mathbb{R}^d$, to which a scalar loss $f_t(z_{t-m+1:t})$, depending on the learner's most recent $m \in \mathbb{N}$ decisions is revealed. Given a convex compact set $\mathcal{K} \subset \mathbb{R}^d$ as the domain of comparators, the regret is measured on the best single point $z \in \mathcal{K}$ over a time horizon of $T \in \mathbb{N}$, formally given by

$$\mathrm{Regret}_T^{\mathcal{K}} = \max_{z \in \mathcal{K}} \mathbb{E} \left[ \sum_{t=m}^{T} f_t(z_{t-m+1:t}) - f_t(z, \cdots, z) \right].$$

In non-stochastic control, the learner's past decisions affect future states through an affine structure, therefore we are interested in BCO-M problems with such structure. This leads to the following structural assumption on the induced loss function with memory. We will show in Lemma 9 that the bandit control problems with the standard regularity conditions satisfy Assumption 5.

**Assumption 5** (Affine memory structure). *At time $t \in \mathbb{N}$, the loss function with memory length $m$ takes the form of $f_t : (\mathbb{R}^d)^m \to \mathbb{R}_+$ given by the following structure*

$$f_t(z_{t-m+1:t}) = \ell_t \left( B_t + \sum_{i=0}^{m-1} G^{[i]} Y_{t-i} z_{t-i} \right), \tag{3}$$

*with parameters $B_t \in \mathbb{R}^n$, $Y_{t-i} \in \mathbb{R}^{p \times d}$, and $G = G^{[0:m-1]}$ a sequence of $m$ matrices where $G^{[i]} \in \mathbb{R}^{n \times p}$ for some $n, p \in \mathbb{N}$. [2] We denote $G_t = \sum_{i=0}^{m-1} G^{[i]} Y_{t-i} \in \mathbb{R}^{n \times d}$, and $H_t = G_t^\top G_t \in \mathbb{R}^{d \times d}$. There exists some $R_H > 0$ such that $\max\{1, \|G_t\|_2, \|Y_t\|_2, \|H_t\|_2\} \leq R_H$. In addition, we assume that $G$ satisfies positive convolution invertibility-modulus, i.e. $\kappa(G) = \inf_{\sum_{n \geq 0} \|u_n\|_2^2 = 1} \sum_{n \geq 0} \|\sum_{i=0}^{n} G^{[i]} u_{n-i}\|_2^2 = \Omega(1)$. We assume that the learner receives $H_t$ every step after they incurred the loss.*

We make two standard curvature and regularity assumptions on each of the instantaneous loss $f_t$, that $f_t$ is strongly-convex and smooth, with its subgradient norm bounded by some constants, following the previous work of Suggala et al. (2024).

**Assumption 6** (Curvature). *Consider a loss function $f_t$ satisfying Assumption 5 and the set*

$$\mathcal{Z}_t = \left\{ B_t + \sum_{i=0}^{m-1} G^{[i]} Y_{t-i} z_{t-i} : z_{t-m+1}, \ldots, z_t \in \mathcal{K} + \mathbb{B}_d \right\} \subset \mathbb{R}^n,$$

*where $B_t, G, Y_{t-m+1:t}$ are the parameters, and $\ell_t : \mathbb{R}^n \to \mathbb{R}_+$ is the function associated with $f_t$ in Assumption 5. We assume that $\ell_t$ is $\alpha_f$-strongly-convex and $\beta_f$-smooth, i.e. $\alpha_f I_n \preceq \nabla^2 \ell_t(z) \preceq \beta_f I_n$, over $\mathcal{Z}_t$, with $0 < \alpha_f \leq 1 \leq \beta_f$ here [3].*

**Assumption 7** (Regularity). *For $d \in \mathbb{N}$, denote $\mathbb{B}_d := \{x \in \mathbb{R}^d \mid \|x\|_2 \leq 1\}$ as the unit ball in $\mathbb{R}^d$. We assume that at time $t$, the with-memory loss function $f_t : (\mathbb{R}^d)^m \to \mathbb{R}_+$ with memory parameter $m \in \mathbb{N}$ obeys the following gradient bounds over $\mathcal{K} + \mathbb{B}_d := \{x + y \mid x \in \mathcal{K}, y \in \mathbb{B}_d\}$:*

$$\|\nabla f_t(z_1, \ldots, z_m)\|_2 \leq G_f, \quad \forall z_1, \ldots, z_m \in \mathcal{K} + \mathbb{B}_d.$$

*Additionally, we assume that $\mathcal{K}$ has Euclidean diameter $D > 0$.*

---

[2] Here we could write $G^{[i]} Y_{t-i}$ with a single parameter, but the current form matches that of control.

[3] Any $\alpha$ strongly convex function is also $\min(1, \alpha)$ strongly convex and similar for $\beta$.

We consider an oblivious adversary model, given by Assumption 8.

**Assumption 8** (Oblivious adversary). *For a BCO-M instance, we assume that the adversary is oblivious. In particular, let $z_t$ denote the algorithm's decision at time $t$, then the loss function $f_t$ chosen by the adversary does not depend on $z_{1:t}$.*

In the definition of regret in BCO-M, we compare the cumulative loss of any algorithm with the best fixed comparator $z \in \mathcal{K}$, evaluated on the induced unary form $f_t(z, \ldots, z)$ of the loss $f_t$. Formally, we have the following definition

**Definition 2** (Induced unary form). $\forall t \in \mathbb{N}$, let $\bar{f}_t : \mathbb{R}^d \to \mathbb{R}_+$ denote the induced unary form of the loss $f_t$, given by $\bar{f}_t(z) = f_t(z, \ldots, z)$. Then, for $f_t$ satisfying Assumption 5, $\bar{f}_t$ admits the structure $\bar{f}_t(z) = \ell_t\left(B_t + \sum_{i=0}^{m-1} G^{[i]} Y_{t-i} z\right)$.

We note that for $f_t$ satisfying Assumption 6, the induced unary form in Definition 2 satisfies a special curvature called $\kappa$-convexity introduced in (Suggala et al., 2024). To avoid confusion with the notations with the bound on the dynamics (Assumption 2), we will call it $\kappa_0$-convexity henceforth.

**Definition 3** ($\kappa_0$-convexity, (Suggala et al., 2024)). A function $f : \mathbb{R}^d \to \mathbb{R}$ is called $\kappa_0$-convex over a domain $\mathcal{K} \subseteq \mathbb{R}^d$ if and only if the following holds: $f$ is convex and twice continuously differentiable, and moreover $\exists c, C > 0$ and a PSD matrix $0 \preceq H \preceq I$ s.t. the Hessian of $f$ at any $z \in \mathcal{K}$ satisfies

$$cH \preceq \nabla^2 f(z) \preceq CH, \quad \frac{C}{c} \leq \kappa_0.$$

The benefit that the loss function exhibits an affine memory dependence is that its induced unary form satisfies $\kappa_0$ convexity for some $\kappa_0 > 0$, summarized the following observation.

**Observation 4** ($\kappa_0$-convexity and gradient bound of the unary loss.). *Consider $f_t : (\mathbb{R}^d)^m \to \mathbb{R}_+$ satisfying Assumption 5, Assumption 6, and Assumption 7. Then, the induced unary form $\bar{f}_t : \mathbb{R}^d \to \mathbb{R}_+$ in Definition 2 satisfies the following two properties:*

1. *($\kappa_0$-convexity) $\bar{f}_t$ is $\kappa_0$-convex with $\kappa_0 = \beta_f/\alpha_f$, $H = H_t$: $\alpha_f H_t \preceq \nabla^2 \bar{f}_t(z) \preceq \beta_f H_t$, where $H_t = G_t^\top G_t$ as in Assumption 5, $\forall z \in \mathbb{R}^d$.*

2. *(Gradient bound) $\|\nabla \bar{f}_t(z)\|_2 \leq G_f \sqrt{m}$, $\forall z \in \mathcal{Z}_t$.*

*Proof of Observation 4.* By our assumption on the affine structure of $f_t$ in Assumption 5, $\forall z \in \mathbb{R}^d$, $\nabla^2 \bar{f}_t(z) = G_t^\top \nabla^2 \ell_t(B_t + G_t z) G_t$. $\kappa_0$-convexity follows from the curvature assumption in Assumption 6. For any $z_1, z_2 \in \mathbb{R}^d$, we have that $|\bar{f}_t(z_1) - \bar{f}_t(z_2)| \leq G_f \|(z_1, \cdots, z_1) - (z_2, \cdots, z_2)\|_2 = G_f \sqrt{m} \|z_1 - z_2\|_2$. $\qquad\square$

### 2.3 Notations

For a positive semidefinite square matrix $H \succeq 0 \in \mathbb{R}^{d \times d}$, define the norm $\|\cdot\|_H$ on $\mathbb{R}^d$ so that $\|v\|_H^2 = v^\top H v$. We write $c_t$ to denote the cost function of the control problem, and $f_t$ to denote the loss function for BCO with memory. For two sets $A, B$, $A + B = \{a + b : a \in A, b \in B\}$. For a set $S$, $|S|$ denotes the cardinality of $S$. We use $\mathbb{B}_d, \mathbb{S}_{d-1}$ to denote the unit ball and unit sphere in $\mathbb{R}^d$, respectively. For $n$ vectors $v_1, \ldots, v_n \in \mathbb{R}^d$, we slightly abuse notation and shorthand as $v_{1:n}$ to denote the concatenated vector $(v_1, \ldots, v_n) \in \mathbb{R}^{nd}$.

## 3  Improved Bandit Convex Optimization with Memory

In this section, we introduce an improved algorithm for the bandit convex optimization with memory problem. Our algorithm incorporates the occasional update idea from (Cassel and Koren, 2020) and the Newton-based update for $\kappa_0$-convex functions from (Suggala et al., 2024), achieving an optimal $\tilde{O}(\sqrt{T})$ regret bound, improving the previously best known $\tilde{O}(T^{2/3})$ result from (Cassel and Koren, 2020). Besides the advancement on BCO-M, this result is the key component of our main result in control. First, we define the BCO-M instance.

**Definition 5** (BCO-M instance). Given $d \in \mathbb{N}$, a BCO-M instance is parametrized by $\mathcal{O} = \{\mathcal{K}, m, \{f_t\}_{t \geq m, t \in \mathbb{N}}\}$, where $\mathcal{K} \subset \mathbb{R}^d$ is a convex compact set; $m$ is the memory length; $f_t : \mathcal{K}^m \to \mathbb{R}_+$ measures the instantaneous loss at time $t$. Given any bandit online learning with memory algorithm $\mathcal{A}^B$, the regret of $\mathcal{A}^B$ on $\mathcal{O}$ w.r.t. $z \in \mathcal{K}$ is given by

$$\text{Regret}_T^{\mathcal{A}^B, z}(\mathcal{O}) = \sum_{t=m}^{T} f_t(z_{t-m+1:t}) - \bar{f}_t(z),$$

where $z_t = z_t^{\mathcal{A}^B}$ is the decision according to $\mathcal{A}^B$ at time $t$. We say a BCO-M instance $\mathcal{O}$ is affine and $(\alpha, \beta, G, D)$-**well-conditioned** if $\mathcal{O}$ satisfies Assumption 5, Assumption 6 with $\alpha_f = \alpha, \beta_f = \beta$, Assumption 7 with $G_f = G, D_f = D$, and the adversary model satisfies Assumption 8.

### 3.1 BCO-M algorithm

We describe Algorithm 1 that runs bandit convex optimization for any BCO-M instance $\mathcal{O} = \{\mathcal{K}, m, \{f_t\}_{t \geq m, t \in \mathbb{N}}\}$. On a high level, our algorithm employs the occasional update idea from (Cassel and Koren, 2020). The decisions are updated at most once every $m$ steps, ensured by the condition $b_t \prod_{i=1}^{m-1} (1 - b_{t-i}) = 1$ (line 8 in Algorithm 1), where $b_t$ is the Bernoulli random variable (partially) deciding whether the algorithm will make an update at the current time step. We write $o$ as the original ideal prediction, $v$ as the random perturbation vector, and $z$ is the actual prediction (as the perturbed $o$) of the algorithm.

---

**Algorithm 1** Improved Bandit Convex Optimization with Affine Memory

---

**Input:** convex compact set $\mathcal{K} \subset \mathbb{R}^d$, step size $\eta > 0$, memory parameter $m$, curvature parameter $\alpha$, time horizon $T$.

1: Initialize: $o_1 = \cdots = o_m \in \mathcal{K}, \tilde{g}_{0:m-1} = \mathbf{0}_d, \hat{A}_{0:m-1} = I, \tau = 1$.
2: Sample $v_t \sim S_{d-1}$ i.i.d. uniformly at random for $t = 1, \ldots, m$.
3: Set $z_t = o_t + \hat{A}_{t-1}^{-\frac{1}{2}} v_t, t = 1, \ldots, m$.
4: Draw $b_t \sim \text{Ber}\left(\frac{1}{m}\right), t = 1, \ldots, m$.
5: **for** $t = m, \ldots, T$ **do**
6:     Play $z_t$, observe $f_t(z_{t-m+1:t})$, receive $H_t = G_t^\top G_t$.
7:     Draw $b_t \sim \text{Ber}\left(\frac{1}{m}\right)$.
8:     **if** $b_t \prod_{i=1}^{m-1}(1 - b_{t-i}) = 1$ **then**
9:         Let $s_\tau = t$.
10:         Update $\hat{A}_t = \hat{A}_{t-1} + \frac{\eta \alpha}{2} H_t$.
11:         Create gradient estimate: $\tilde{g}_t = df_t(z_{t-m+1:t})\hat{A}_{t-1}^{\frac{1}{2}} v_t \in \mathbb{R}^d$.
12:         Update $o_{t+1} = \prod_{\mathcal{K}}^{\hat{A}_{s_{\tau-1}}} \left[ o_t - \eta \hat{A}_{s_{\tau-1}}^{-1} \tilde{g}_{s_{\tau-1}} \right]$.
13:         Sample $v_{t+1} \sim S_{d-1}$ uniformly at random, independent of previous steps.
14:         Set $z_{t+1} = o_{t+1} + \hat{A}_t^{-\frac{1}{2}} v_{t+1}$.
15:         $\tau \leftarrow \tau + 1$.
16:     **else**
17:         Set $o_{t+1} = o_t, v_{t+1} = v_t, z_{t+1} = z_t, \hat{A}_t = \hat{A}_{t-1}, \tilde{g}_t = \tilde{g}_{t-1}$.
18:     **end if**
19: **end for**

---

Such occasional update essentially reduces the BCO-M problem to a new no-memory BCO problem, whose equivalence will be shown later (see Appendix A.2). Consistent with the notation used in (Cassel and Koren, 2020), we denote the following set

$$S := \{t \in [T] : z_{t+1} \neq z_t\} \tag{4}$$

to be the set of time steps where the algorithm updates its decision. We readily have $|S| \leq \frac{T}{m}$. Moreover, we have

$$f_t(z_{t-m+1:t}) = \bar{f}_t(z_{t-m+1}) = \bar{f}_t(z_t), \quad \forall t \in S, \tag{5}$$

since $t \in S$ implies $b_{t-m+1} = \cdots = b_{t-1} = 0$. Thus, whenever Algorithm 1 updates, it effectively updates with function value $\bar{f}_t(z_t)$.

The occasional update alone only gives a sub-optimal $\tilde{O}(T^{2/3})$ regret as shown in (Cassel and Koren, 2020). To achieve the optimal $\tilde{O}(\sqrt{T})$ regret, we use a Newton-based update to exploit the $\kappa_0$-convexity of general strongly-convex smooth functions, recently introduced by (Suggala et al., 2024).

For this improvement we require the knowledge of a Hessian estimator $H_t$ as in Assumption Assumption 5. This doesn't hold in BCO in general, but we will show in the next section that the control problem indeed satisfies this assumption: thanks to $\kappa_0$-convexity, in the control problem $H_t$ can be constructed from the knowledge of system, instead of knowledge of loss which would typically incur a huge variance term.

The regret guarantee of Algorithm 1 is given by the following theorem.

**Theorem 6** (BCO-M regret guarantee). *Given an $(\alpha, \beta, G, D)$-well-conditioned BCO-M instance $\mathcal{O} = \{\mathcal{K}, m, \{f_t\}_{t \geq m, t \in \mathbb{N}}\}$ with $m = \mathrm{poly}(\log T)$ and $G, D = \tilde{O}(1)$ (Definition 5), let $(\mathcal{K}, \eta = \Theta(1/\sqrt{T}), m, \alpha, T)$ be the input to Algorithm 1. Algorithm 1 guarantees that*

$$\max_{z \in \mathcal{K}} \mathbb{E}\left[ Regret_T^{Algorithm\ 1, z}(\mathcal{O}) \right] \leq \tilde{O}\left( \frac{\beta}{\alpha} GD\sqrt{T} \right),$$

*where $\tilde{O}(\cdot)$ hides all universal constants and logarithmic dependence in $T$.*

Theorem 6 is the first algorithm to achieve the optimal $\tilde{O}(\sqrt{T})$ regret bound for the BCO-M problem with general smooth convex loss. Compared with the $\tilde{O}(T^{2/3})$ bound from (Cassel and Koren, 2020), our improvement exploits the new assumptions on the strong convexity of loss and the affine structure of memory. We notice that the $\tilde{O}(\sqrt{T})$ regret bound of (Suggala et al., 2024) requires the additional quadratic loss assumption and uses the special structure of quadratic functions to construct low-biased gradient estimators for the unary form of losses. Algorithm 1 and its guarantee in Theorem 6 thus improve upon both of these previous results. The proof of Theorem 6 is lengthy, therefore we leave it to the appendix and include a sketch here.

### 3.2 Proof Sketch

The theorem is proven via reduction to a no-memory BCO algorithm with Newton-based updates.

**Step 1: regret of the base algorithm.** We devise a new BCO algorithm with two new ingredients different from standard algorithms. First, our algorithm adopts Newton-based updates which require estimating Hessians. In Assumption 5 we assume "free" access to a Hessian estimator $H_t$, which holds in the control setting by utilizing the $\kappa_0$-convexity property.

Second, we incorporate a new delay mechanism to decorrelate neighboring iterates such that $o_t$ is independent of recent perturbation vectors, which plays a crucial role in bounding the expectation of moving cost. Our base BCO algorithm is then shown to have an $\tilde{O}(\sqrt{T})$ regret bound (see Lemma 11).

**Step 2: reduction to the base algorithm.** We show that the regret of Algorithm 1 can be controlled by the regret of the base algorithm plus a moving cost term. By the design of Algorithm 1, it updates a univariate loss function at most once every $m$ steps. If all the $z_t$ across these $m$ steps are the same, **Regret**(Algorithm 1) will be exactly the same as $m \times$ **Regret**(base algorithm). When $z_t$ are different, we suffer an additional moving cost $\mathbb{E}\left[ \sum_{t=m}^T f_t(z_{t-m+1:t}) - \bar{f}_t(z_{t-m+1}) \right]$, which can be bounded by the assumption on the gradient of $\bar{f}_t$ if $z_t$ changes slowly.

**Step 3: bounding the moving cost.** We partition the moving cost $f_t(z_{t-m+1:t}) - \bar{f}_t(z_{t-m+1})$ into three terms

$$\underbrace{f_t(z_{t-m+1:t}) - f_t(o_{t-m+1:t})}_{(a)} + \underbrace{f_t(o_{t-m+1:t}) - \bar{f}_t(o_{t-m+1})}_{(b)} + \underbrace{\bar{f}_t(o_{t-m+1}) - \bar{f}_t(z_{t-m+1})}_{(c)}.$$

Here, (a), (c) can be seen as perturbation loss suffered by the algorithm during exploration. (b) is the moving cost determined by the stability of the algorithm's neighboring iterates.

For the three terms in the above equation, (c) is bounded by Jensen's inequality using the convexity of the unary form of loss. We bound (a) using the curvature assumptions and the affine memory structure of $f_t$, where we also make use of the delayed updates to decorrelate neighboring iterates for technical reasons. (b) is bounded by the Lipschitzness of $f_t$ and the distance between neighboring iterates. Theorem 6 is reached by putting the three steps together.

## 4   Optimal Regret for Bandit Non-stochastic Control

In this section, we show how to achieve optimal regret for the bandit non-stochastic control problem with a partially observable LTI dynamical system as described in Eq. (1) for strongly-convex smooth loss, by a reduction to the BCO-M algorithm (Algorithm 1) we devise in the previous section. Previously, the best known regret bound for this problem was $\tilde{O}(T^{2/3})$ from (Cassel and Koren, 2020), which is also rooted in a reduction to BCO-M. We use a similar reduction, obtaining a better bound thanks to the improved regret bound of BCO-M (Theorem 6). We first give the formal definition of the control problem.

**Definition 7** (Bandit non-stochastic control). A bandit non-stochastic control problem of a partially observable LDS is parametrized by a tuple $\mathcal{L} = (A, B, C, x_1, (w_t)_{t \in \mathbb{N}}, (e_t)_{t \in \mathbb{N}}, (c_t)_{t \in \mathbb{N}}, \Pi)$, where $A, B, C$ and $(w_t)_{t \in \mathbb{N}}, (e_t)_{t \in \mathbb{N}}$ are the dynamics and perturbations in the LDS (Eq. (1)) with initial state $x_1$ (we assume $x_1 = 0$ without loss of generality henceforth); $c_t$ measures the instantaneous cost at time $t$; $\Pi$ is the comparator control policy class. Given any bandit non-stochastic control algorithm $\mathcal{A}$, the regret of $\mathcal{A}^{\mathrm{NC}}$ on $\mathcal{L}$ over a time horizon $T \in \mathbb{N}$ is given by

$$\mathrm{Regret}_T^{\mathcal{A}^{\mathrm{NC}}}(\mathcal{L}, \Pi) = \sum_{t=1}^T c_t(y_t^{\mathcal{A}^{\mathrm{NC}}}, u_t^{\mathcal{A}^{\mathrm{NC}}}) - \min_{\pi \in \Pi} \sum_{t=1}^T c_t(y_t^\pi, u_t^\pi),$$

where $(y_t^{\mathcal{A}^{\mathrm{NC}}}, u_t^{\mathcal{A}^{\mathrm{NC}}}), (y_t^\pi, u_t^\pi)$ are the observation, control pair at time $t$ following the trajectory of $\mathcal{A}^{\mathrm{NC}}$ and $\pi$, respectively. We say that a bandit non-stochastic control problem $\mathcal{L}$ is $(\alpha, \beta, G, \kappa_{\mathbf{sys}}, \kappa, \gamma, R_{\mathbf{w},\mathbf{e}}, \mathcal{Y}, \mathcal{U})$-**well-conditioned** if $\mathcal{L}$ and $\Pi$ satisfy Assumption 1 with $\alpha_c = \alpha, \beta_c = \beta, G_c = G$ over some centered bounded convex domain $\mathcal{Y} \times \mathcal{U} \subset \mathbb{R}^{d_y + d_u}$, Assumption 2 with $\kappa_{\mathbf{sys}}, \kappa, \gamma$, Assumption 3 with $R_{\mathbf{w},\mathbf{e}}$, and Assumption 4.

The reduction from partially observable bandit non-stochastic control to BCO-M consists of two main steps. The first step is to construct the would-be signal $y_t(K)$ from the observation $y_t$, where $y_t(K)$ is used for updating but not directly observed by the controller. $y_t(K)$ is computed by using the Markov operator of the LDS (Simchowitz et al., 2020). The second step is a standard black-box reduction from control to BCO-M, which uses the strong stability of the system. Reduction is formalized in the following definition.

**Definition 8** (Approximation). A bandit non-stochastic control instance (Definition 7) $\mathcal{L}$ with some convex comparator class $\Pi$ over a time horizon $T \in \mathbb{N}$ is said to be $\varepsilon$-**approximated** ($\varepsilon > 0$) by a BCO-M instance (Definition 5) $\mathcal{O}$ with domain $\mathcal{K} = \Pi$, if the existence of a BCO-M algorithm $\mathcal{A}^B$ implies the existence of a bandit non-stochastic control algorithm $\mathcal{A}^{\mathrm{NC}}$ satisfying

$$\mathbb{E}\left[\mathrm{Regret}_T^{\mathcal{A}^{\mathrm{NC}}, \mathcal{K}}(\mathcal{L}, \Pi)\right] \leq \mathbb{E}\left[\mathrm{Regret}_T^{\mathcal{A}^B, \mathcal{K}}(\mathcal{O})\right] + \varepsilon T.$$

The formal guarantee of the reduction is given below.

**Lemma 9** (Control reduction). *Every instance of $(\alpha_c, \beta_c, G_c, \kappa_{\mathbf{sys}}, \kappa, \gamma, R_{\mathbf{w},\mathbf{e}}, \mathcal{Y}, \mathcal{U})$-well-conditioned bandit non-stochastic control problem $\mathcal{L}$ over the ball $\mathcal{Y} \times \mathcal{U} \subset \mathbb{R}^{d_y + d_u}$ of radius $R$ [4] with comparator class $\Pi = \mathcal{M}(m, R_{\mathcal{M}})$ (Definition 1) is $2G_f D m T^{-1}$-approximated by a $(\alpha_f, \beta_f, G_f, D)$-well-conditioned BCO-M instance $\mathcal{O}$ with*

$$\alpha_f = \alpha_c, \quad \beta_f = \beta_c, \quad D = \sqrt{m \max\{d_u, d_y\}} R_{\mathcal{M}}, \quad G_f = \frac{4096\sqrt{m} G_c R_{w,e}^2 R_{\mathcal{M}}^2 d_x^{2.5} \kappa^3 \kappa_{\mathbf{sys}}^8}{\gamma^5},$$

*provided that $m = \Theta(\log T / \log(1/(1-\gamma)))$.*

---

[4]See Appendix B, $R = R_{w,e}\left(1 + \frac{\sqrt{d_x}\kappa_{\mathbf{sys}}}{\gamma}\right)\left(\sqrt{1+\kappa^2} + R_{\mathcal{M}}\left(1 + \frac{\sqrt{d_x(1+\kappa^2)}\kappa_{\mathbf{sys}}^2}{\gamma}\right)\right) = O(1)$.

Lemma 9 implies that every well-conditioned bandit non-stochastic control problem can be reduced to a well-conditioned BCO-M instance, whose parameters are polynomial in those of the control problem. As a result, any regret bound for BCO-M will directly transfer to the control problem (at the expense of polynomial dependence on the system parameters). In particular, combining Theorem 6 and Lemma 9, we obtain an optimal $\tilde{O}(\sqrt{T})$ regret bound for the bandit non-stochastic control with strongly-convex smooth cost.

**Theorem 10** (Bandit non-stochastic control regret guarantee). *Given an $(\alpha, \beta, G, \kappa_{sys}, \kappa, \gamma, R_{\mathbf{w}, \mathbf{e}}, \mathcal{Y}, \mathcal{U})$-well-conditioned bandit non-stochastic control instance $\mathcal{L}$ over the ball $R\mathbb{B}_{d_y + d_u} \subset \mathbb{R}^{d_y + d_u}$ for the same $R$ as in Lemma 9, with $G, \kappa_{sys}, \kappa, \gamma, R_{\mathbf{w}, \mathbf{e}}, d_x, d_y, d_u = \tilde{O}(1)$[5], let $(\Pi, \mathcal{K}, \eta, m, T, G, K, \alpha)$ be the input to Algorithm 2 with $m = \text{poly}(\log T)$. Algorithm 2 guarantees that*

$$\max_{z \in \mathcal{K}} \mathbb{E}\left[ Regret_T^{Algorithm\ 2, z}(\mathcal{O}) \right] \leq \tilde{O}\left( \frac{\beta}{\alpha} \sqrt{T} \right),$$

*where $\tilde{O}(\cdot)$ hides all universal constants and logarithmic dependence in $T$.*

Theorem 10 strictly improves Theorem 8 of (Cassel and Koren, 2020) and Theorem 5 of (Suggala et al., 2024), in the sense that our result achieves the optimal regret with fewer assumptions: all three results achieve the same $\tilde{O}\left( \frac{\beta}{\alpha} \sqrt{T} \right)$ regret bound, however (Cassel and Koren, 2020) requires the additional assumption that perturbations are stochastic, while (Suggala et al., 2024) requires the additional assumption that costs are quadratic.

---

**Algorithm 2** Improved Bandit Non-stochastic Control

---

**Input:** Step size $\eta > 0$, memory parameter $m$, DRC policy class $\mathcal{M}(m, R_{\mathcal{M}})$, time horizon $T$, system dynamics, $(\kappa, \gamma)$-strongly stabilizing linear policy $K$, strong convexity parameter $\alpha > 0$.

1: Let $\mathcal{A}^{\mathrm{B}}$ be an instance of Algorithm 1 with inputs $(\mathcal{K} = \mathcal{M}(m, R_{\mathcal{M}}), \eta, m, \alpha, T)$.
2: Initialize: $M_1^{[j]} = \cdots = M_m^{[j]} = 0_{d_u \times d_y}, \forall j \in [m]$. $\tilde{g}_0 = \cdots = \tilde{g}_{m-1} = 0_{md_u d_y}$, $\hat{A}_0 = \hat{A}_{m-1} = mI_{md_u d_y \times md_u d_y}$.
3: Initialize $\mathcal{A}^{\mathrm{B}}$ for $t = 1, \cdots, m-1$ (lines 1-4 in Algorithm 1).
4: Sample $\xi_t \sim S^{md_u d_y - 1}$ i.i.d. uniformly at random for $t = 1, \ldots, m$.
5: Play control $u_t = Ky_t$, incur cost $c_t(y_t, u_t)$ for $t \in [m]$.
6: **for** $t = m, \ldots, T$ **do**
7:    Play control $u_t^{M_t} = Ky_t + \sum_{j=0}^{m-1} M_t^{[j]} y_{t-j}(K)$, incur cost $c_t(y_t, u_t)$.
8:    Let $f_t : (\mathcal{M}(m, R_{\mathcal{M}}))^m \to \mathbb{R}$ be the induced with-memory loss function via reduction in Lemma 9 and $H_t$ be the associated Hessian estimator in Assumption 5.
9:    Update $M_{t+1} \leftarrow \mathcal{A}^{\mathrm{B}}(M_t, \{f_s\}_{s=m}^t, \{H_s\}_{s=m}^t)$.
10: **end for**

---

# 5  Conclusion

In this paper, we devise an algorithm with an $\tilde{O}(\sqrt{T})$ regret bound for the bandit non-stochastic control problem with adversarial strongly-convex smooth cost functions. This is the first result with optimal regret that simultaneously breaks the three assumptions of LQC (1) stochastic perturbation (2) full-information feedback (3) quadratic cost. Our control algorithm is built upon an improved algorithm for BCO-M, which may be of independent interest.

As a preliminary step to address the question of a general control theory for LTI, our result comes with limitations and potential future research directions. Currently, the improvement over the $\tilde{O}(T^{2/3})$ regret by (Cassel and Koren, 2020) is made under the additional assumption of strong convexity. It's unclear whether this assumption is necessary for an $\tilde{O}(\sqrt{T})$ regret, we leave determining the minimal assumption on the cost functions for optimal regret as an open question.

---

[5]The assumption that the system parameters are of order $\tilde{O}(1)$ is consistent with prior works (Hazan and Singh, 2022).

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

# Contents

# A Proof of Theorem 6

The proof is reduction-based. It consists of three parts

1. We first consider an algorithm for the no-memory BCO problem, which uses Newton-based updates and a novel delay mechanism. This algorithm will serve as the base algorithm.

2. We then show that the regret of our main algorithm 1 can be bounded by the regret of the base algorithm plus a moving cost, via an improved analysis on $\kappa_0$-convexity.

3. Finally, we bound the moving cost and then put every pieces together.

## A.1 Base No-Memory BCO Algorithm and Guarantees

In this section, we prove Lemma 11, which establishes the regret guarantee for the base algorithm used by Algorithm 1. The base no-memory algorithm is a variant of the Newton-based (improper) BCO algorithm considered in Algorithm 1 of (Suggala et al., 2024). The main differences are:

1. Algorithm 1 in (Suggala et al., 2024) constructs Hessian estimators from the bandit feedback. Algorithm 3 doesn't require Hessian estimators, since the $H_t$ determining $\kappa_0$-convexity is given by the system instead of loss in bandit control problems, and thus it's computable by the learner given the system parameters.

2. Algorithm 3 introduces an additional delay mechanism (specified by the delay parameter $d_0 \in \mathbb{N}$) to decorrelate neighboring iterates, a necessary ingredient to later analysis.

---

**Algorithm 3** Simple BCO-with-delay

---

**Input:** convex compact set $\mathcal{K} \subset \mathbb{R}^d$, step size $\eta > 0$, delay parameter $d_0 \in \mathbb{N}$, curvature parameter $\alpha > 0$, time horizon $T \in \mathbb{N}$.

1: Initialize: $o_1 \in \mathcal{K}$, $\hat{A}_0 = I_{d \times d}$. $\hat{A}_t = 0_{d \times d}, \forall t < 0$. $\tilde{g}_t = 0, \forall t \leq 0$.
2: Sample $v_1 \sim S_{d-1}$ uniformly at random. Set $z_1 = o_1 + \hat{A}_0^{-\frac{1}{2}} v_1$.
3: **for** $t = 1, \ldots, T$ **do**
4:     Play $z_t$, observe $\bar{f}_t(z_t)$, receive $H_t$.
5:     Update $\hat{A}_t = \hat{A}_{t-1} + \frac{\eta \alpha}{2} H_t$.
6:     Create gradient estimate: $\tilde{g}_t = d\bar{f}_t(z_t)\hat{A}_{t-1}^{\frac{1}{2}} v_t \in \mathbb{R}^d$.
7:     Update $o_{t+1} = \prod_{\mathcal{K}}^{\hat{A}_{t-d_0+1}} \left[ o_t - \eta \hat{A}_{t-d_0+1}^{-1} \tilde{g}_{t-d_0+1} \right]$.
8:     Sample $v_{t+1} \sim S_{d-1}$ uniformly at random, independent of previous steps.
9:     Set $z_{t+1} = o_{t+1} + \hat{A}_t^{-\frac{1}{2}} v_{t+1}$.
10: **end for**

---

**Lemma 11** (Base BCO regret guarantee). *Suppose that the sequence of loss functions $\{\bar{f}_t\}_{t=1}^T$ and the convex compact set $\mathcal{K}$ satisfy the conditions in Assumption 7, Assumption 8, and the properties in Observation 4, and $\max\{1, \|H_t\|_2\} \leq R_H, \forall t$. With $\alpha = \alpha_f$ in the first condition in Observation 4, $(\eta, d_0)$ satisfying $d_0 \leq 2/(\eta \alpha R_H)$, Algorithm 3 run with inputs $(\mathcal{K}, \eta, d_0, \alpha, T)$ satisfies the following regret guarantee: $\forall o \in \mathcal{K}$,*

$$\mathbb{E}\left[ \sum_{t=1}^T \bar{f}_t(z_t) - \bar{f}_t(o) \right] \leq \frac{2\beta d}{\eta \alpha} \log(\eta R_H T + 1) + 2d_0 GD + \frac{D^2 d_0 R_H}{2\eta} + 3\eta d_0 d^2 G^2 D^2 R_H T.$$

*In particular, by choosing $\eta = \Theta(1/\sqrt{T})$ and $d_0 = \tilde{\Theta}(1)$, the above regret is of order $\tilde{O}\left( \frac{\beta}{\alpha} \sqrt{T} \right)$.*

*Proof of Lemma 11.* First, note that by the condition on $\eta$ and $d_0$, we have that $\hat{A}_t \preceq 2\hat{A}_{t-i}, \forall i \leq d_0$. To see this, it is equivalent to proving any PSD matrix $H$ with $\lambda_{\max}(H) \leq 1$ satisfies that $H \preceq I$, which follows from the fact that $x^\top (I - H)x \geq \|x\|_2^2 - \lambda_{max}\|x\|_2^2 \geq 0, \forall x$.

By the convexity and curvature assumption on $\bar{f}_t$ described by the conditions in Observation 4, we have

$$\mathbb{E}\left[\sum_{t=1}^{T}\bar{f}_t(z_t) - \bar{f}_t(o_t)\right] \le \sum_{t=1}^{T}\mathbb{E}[\nabla\bar{f}_t(o_t)^\top \hat{A}_{t-1}^{-\frac{1}{2}}v_t] + \frac{\beta}{2}\sum_{t=1}^{T}\mathbb{E}[v_t^\top \hat{A}_{t-1}^{-\frac{1}{2}}H_t\hat{A}_{t-1}^{-\frac{1}{2}}v_t],$$

where the first-order term equals 0 since $v_t$ is drawn independently of $o_t$, $\bar{f}_t$, $\hat{A}_{t-1}$, and $\mathbb{E}[v_t] = 0$. We can further bound the second order term by

$$\mathbb{E}\left[\sum_{t=1}^{T}\bar{f}_t(z_t) - \bar{f}_t(o_t)\right] \le \frac{\beta}{2}\sum_{t=1}^{T}\mathbb{E}[\hat{A}_{t-1}^{-1}\cdot H_t] \le \frac{2\beta}{\eta\alpha_f}\sum_{t=1}^{T}\mathbb{E}[\hat{A}_t^{-1}\cdot(\hat{A}_t - \hat{A}_{t-1})],$$

where the first step follows from $\mathbb{E}[v^\top Av|A] = A\cdot\mathbb{E}[vv^\top]$ and the cyclic property of trace, and the last step follows from the definition of $\hat{A}_t$, the stability condition that $\hat{A}_t \preceq 2\hat{A}_{t-1}$, and $\alpha = \alpha_f$. Using standard inequalities on log determinant in Newton step analysis (Hazan et al., 2007), we have

$$\mathbb{E}\left[\sum_{t=1}^{T}\bar{f}_t(z_t) - \bar{f}_t(o_t)\right] \le \frac{2\beta}{\eta\alpha_f}\log\frac{\det(\hat{A}_T)}{\det(\hat{A}_0)} \le \frac{2\beta d}{\eta\alpha_f}\log\left(\frac{\eta\alpha_f R_H T}{2} + 1\right).$$

Moreover, the curvature assumption on $\bar{f}_t$ also implies

$$\mathbb{E}\left[\sum_{t=1}^{T}\bar{f}_t(o_t) - \bar{f}_t(o)\right] \le \sum_{t=1}^{T}\mathbb{E}[\nabla\bar{f}_t(o_t)^\top(o_t - o)] - \frac{\alpha_f}{2}\sum_{t=1}^{T}\mathbb{E}[\|o_t - o\|_{H_t}^2]$$

$$= \sum_{t=1}^{T}\mathbb{E}[\tilde{g}_t^\top(o_t - o)] - \frac{\alpha_f}{2}\sum_{t=1}^{T}\mathbb{E}[\|o_t - o\|_{H_t}^2], \tag{6}$$

where the second step follows from $\mathbb{E}[\tilde{g}_t \mid o_t] = \nabla\bar{f}_t(o_t)$ by Stoke's theorem. By the projection step in Line 7 of Algorithm 3, we have

$$\|o_t - o\|_{\hat{A}_{t-d_0}}^2$$

$$\le \|o_{t-1} - o - \eta\hat{A}_{t-d_0}^{-1}\tilde{g}_{t-d_0}\|_{\hat{A}_{t-d_0}}^2$$

$$= \|o_{t-1} - o\|_{\hat{A}_{t-d_0-1}}^2 + \frac{1}{2}\|o_{t-1} - o\|_{\eta\alpha_f H_{t-d_0}}^2 - 2\eta\tilde{g}_{t-d_0}^\top(o_{t-1} - o) + \eta^2\|\tilde{g}_{t-d_0}\|_{\hat{A}_{t-d_0}^{-1}}^2$$

$$\le \|o_{t-1} - o\|_{\hat{A}_{t-d_0-1}}^2 + \|o_{t-d_0} - o\|_{\eta\alpha_f H_{t-d_0}}^2 + \|o_{t-d_0} - o_{t-1}\|_{\eta\alpha_f H_{t-d_0}}^2 - 2\eta\tilde{g}_{t-d_0}^\top(o_{t-d_0} - o)$$

$$+ 2\eta\tilde{g}_{t-d_0}^\top(o_{t-d_0} - o_{t-1}) + \eta^2\|\tilde{g}_{t-d_0}\|_{\hat{A}_{t-d_0}^{-1}}^2,$$

where the last inequality follows from that $\forall H \succeq 0$, $\|x + y\|_H^2 \le 2(\|x\|_H^2 + \|y\|_H^2)$.

Rearranging, we have

$$\tilde{g}_{t-d_0}^\top(o_{t-d_0} - o) - \frac{\alpha_f}{2}\|o_{t-d_0} - o\|_{H_{t-d_0}}^2$$

$$\le \frac{\|o_{t-1} - o\|_{\hat{A}_{t-d_0-1}}^2 - \|o_t - o\|_{\hat{A}_{t-d_0}}^2}{2\eta} + \frac{\alpha_f}{2}\|o_{t-d_0} - o_{t-1}\|_{H_{t-d_0}}^2 + \tilde{g}_{t-d_0}^\top(o_{t-d_0} - o_{t-1}) + \frac{\eta}{2}\|\tilde{g}_{t-d_0}\|_{\hat{A}_{t-d_0}^{-1}}^2.$$

Note that $\forall t$,

$$\|\tilde{g}_t\|_{\hat{A}_t^{-1}}^2 \le (dGD)^2 v_t^\top \hat{A}_{t-1}^{\frac{1}{2}}\hat{A}_t^{-1}\hat{A}_{t-1}^{\frac{1}{2}}v_t \le (dGD)^2,$$

$$\|o_t - o_{t-1}\|_2 \le \|o_t - o_{t-1}\|_{\hat{A}_{t-d_0}} \le \eta\|\tilde{g}_{t-d_0}\|_{\hat{A}_{t-d_0}^{-1}} \le \eta dGD,$$

where the second inequality in the second bound follows from the update rule in Line 7 of Algorithm 3, which gives $\|o_t - o_{t-1}\|_{\hat{A}_{t-d_0}}^2 \le \|\eta\hat{A}_{t-d_0}^{-1}\tilde{g}_{t-d_0}\|_{\hat{A}_{t-d_0}}^2 = \eta^2\|\tilde{g}_{t-d_0}\|_{\hat{A}_{t-d_0}^{-1}}^2$. Thus, we can further bound

$$\frac{\alpha_f}{2}\|o_{t-d_0} - o_{t-1}\|_{H_{t-d_0}}^2 \le \frac{\alpha_f R_H d_0}{2}\sum_{i=0}^{d_0}\|o_{t-d_0+i} - o_{t-d_0+i+1}\|_2^2 \le \frac{\eta^2\alpha_f d_0 d^2 G^2 D^2 R_H}{2},$$

$$\tilde{g}_{t-d_0}^\top(o_{t-d_0} - o_{t-1}) \le \|\tilde{g}_{t-d_0}\|_{\hat{A}_{t-d_0}^{-1}}\|o_{t-d_0} - o_{t-1}\|_{\hat{A}_{t-d_0}} \le 2\eta d_0 d^2 G^2 D^2.$$

Combining both and using that $\hat{A}_t \preceq 2\hat{A}_{t-i}, \forall i \leq d_0$, we have that

$$\sum_{t=d_0}^{T-d_0} \tilde{g}_t^\top (o_t - o) - \frac{\alpha_f}{2} \|o_t - o\|_{H_t}^2$$

$$= \sum_{t=2d_0}^{T} \tilde{g}_{t-d_0}^\top (o_{t-d_0} - o) - \frac{\alpha_f}{2} \|o_{t-d_0} - o\|_{H_{t-d_0}}^2$$

$$\leq \frac{1}{2\eta} \|o_{2d_0-1} - o\|_{\hat{A}_{d_0-1}}^2 + \eta T \left( \frac{\alpha_f d_0 d^2 G^2 D^2 R_H}{2} + 2d_0 d^2 G^2 D^2 + \frac{d^2 G^2 D^2}{2} \right)$$

$$\leq \frac{D^2 d_0 R_H}{2\eta} + 3\eta d_0 d^2 G^2 D^2 R_H T.$$

As a result,

$$\mathbb{E}\left[\sum_{t=1}^{T} \bar{f}_t(o_t) - \bar{f}_t(o)\right] \leq 2d_0 GD + \frac{D^2 d_0 R_H}{2\eta} + 3\eta d_0 d^2 G^2 D^2 R_H T.$$

Combining, we have

$$\mathbb{E}\left[\sum_{t=1}^{T} \bar{f}_t(z_t) - \bar{f}_t(o)\right] \leq \frac{2\beta d}{\eta\alpha} \log(\eta R_H T + 1) + 2d_0 GD + \frac{D^2 d_0 R_H}{2\eta} + 3\eta d_0 d^2 G^2 D^2 R_H T.$$

$\square$

## A.2 Reduction

In this section, we prove Lemma 12, which states that the regret of BCO-M algorithm (Algorithm 1) can be related to the regret guarantee of the base no-memory BCO algorithm (Algorithm 3).

We follow the proof idea of (Cassel and Koren, 2020). In the previous section we have proved an $\tilde{O}(\sqrt{T})$ regret bound for a "no-memory" base BCO algorithm. Our main Algorithm 1, however, has memory dependence. When $z_t$ is changing slowly, Algorithm 1 is approximately a "no-memory" algorithm, and we can do the following reduction: if all the $z_t$ across these steps are the same, **Regret**(Algorithm 1) will be exactly the same as $m \times$ **Regret**(base algorithm).

In this part, we show that when $z_t$ are different, we only suffer an additional moving cost

$$\mathbb{E}\left[\sum_{t=m}^{T} f_t(z_{t-m+1:t}) - \bar{f}_t(z_{t-m+1})\right],$$

which we will bound in Appendix A.3.

**Lemma 12** (Lemma 11, (Cassel and Koren, 2020)). *Let $(\mathcal{K}, \eta, m, T)$ be the input for Algorithm 1. Suppose that the loss functions $\{f_t\}_{t=m}^{T}$ and the convex compact set $\mathcal{K}$ satisfy Assumption 5, Assumption 6, Assumption 7, and Assumption 8. Then, the regret of Algorithm 1 with respect to any $o \in \mathcal{K}$ is upper bounded by*

$$\mathbb{E}\left[\sum_{t=m}^{T} f_t(z_{t-m+1:t}) - \bar{f}_t(o)\right] \leq 3m \cdot R_{Algorithm\ 3}\left(\frac{T}{m}\right) + \mathbb{E}\left[\sum_{t=m}^{T} f_t(z_{t-m+1:t}) - \bar{f}_t(z_{t-m+1})\right],$$

*where $R_{Algorithm\ 3}\left(\frac{T}{m}\right)$ is the regret upper bound obtained in Lemma 11 with time horizon $\frac{T}{m}$ and $d_0 = 2$.*

*Proof of Lemma 12.* First, note that by assumption, the conditions in Lemma 11 are satisfied for the induced unary forms $\{\bar{f}_t\}_{t=m}^{T}$. The argument follows by reducing Algorithm 1 to Algorithm 3.

Denote $\chi_t = b_t \prod_{i=1}^{m-1}(1 - b_{t-i})$ as the indicator of whether the algorithm gets updated during round $t$. Recall the definition of $S = \{t \in [T] : \chi_t = 1\}$ in Eq. (4). The algorithm updates during round $t$ if $t \in S$. For any $t \in S$, we have that $\chi_{t-1} = \cdots = \chi_{t-m+1} = 0$. Therefore, we have $z_{t-m+1} = \cdots =$

$z_t$ by design of Algorithm 1. Thus, we have that $f_t(z_{t-m+1:t}) = \bar{f}_t(z_{t-m+1}) = \bar{f}_t(z_t)$. Therefore, constrained to the time steps $t \in S$, Algorithm 1 is essentially running Algorithm 3 with a delay parameter $d_0 = 2$ (since in Line 12 of Algorithm 1, we update with the gradient information at time $s_{\tau-1}$ during round $t$). Note that since $|S| \leq \frac{T}{m}$ (Algorithm 1 updates at most once every $m$ steps), we have that $\forall o \in \mathcal{K}$,

$$\mathbb{E}_{b_{1:T}, \{v_t\}_{t \in S}} \left[ \sum_{t \in S} \bar{f}_t(z_t) - \sum_{t \in S} \bar{f}_t(o) \right] \leq \mathbb{E}_{b_{1:T}} \left[ R_{Algorithm \ 3}(|S|) \right] \leq R_{Algorithm \ 3}\left( \frac{T}{m} \right).$$

It is left to relate the quantity on the left hand side of the above expression to the regret of Algorithm 1. Since $\chi_t \overset{D}{=} \chi_m$ (the two random variables equal in distribution), $\forall t$, we have $\mathbb{E}[\chi_t] = \mathbb{E}[\chi_m]$. For any fixed $o \in \mathcal{K}$, we have

$$\mathbb{E}_{b_{1:T}, v_{1:T}} \left[ \sum_{t \in S} \bar{f}_t(o) \right] = \mathbb{E}_{b_{1:T}, v_{1:T}} \left[ \sum_{t=1}^{T} \bar{f}_t(o) \cdot \chi_t \right] = \mathbb{E}[\chi_m] \cdot \mathbb{E} \left[ \sum_{t=m}^{T} \bar{f}_t(o) \right]. \tag{7}$$

For $t \in \mathbb{N}$, denote $\mathcal{F}_t$ to be the $\sigma$-algebra generated by the randomness of Algorithm 1 through sampling $b_s, v_s$ up to time $t$, i.e. $\mathcal{F}_t = \sigma(\{b_s, v_s\}_{s=1}^t)$. Since $z_t = z_{t-m+1}$ whenever $\chi_t = 1$, $b_{t-m+1:t}$ are drawn independently of $z_{t-m+1}$ and $\chi_t$ is independent of $\mathcal{F}_{t-m}$ by definition of $\chi_t$, we have

$$\mathbb{E}_{b_{1:T}, v_{1:T}} \left[ \sum_{t \in S} \bar{f}_t(z_t) \right] = \mathbb{E}_{b_{1:T}, v_{1:T}} \left[ \sum_{t=m}^{T} \bar{f}_t(z_{t-m+1}) \cdot \chi_t \right]$$

$$= \mathbb{E}_{b_{1:T}, v_{1:T}} \left[ \sum_{t=m}^{T} \bar{f}_t(z_{t-m+1}) \cdot \chi_t \mid \mathcal{F}_{t-m}, v_{t-m+1} \right]$$

$$= \mathbb{E}[\chi_m] \cdot \mathbb{E} \left[ \sum_{t=m}^{T} \bar{f}_t(z_{t-m+1}) \right]. \tag{8}$$

Together, Eq. (7) and Eq. (8) give that

$$\mathbb{E}_{b_{1:T}, \{v_t\}_{t \in S}} \left[ \sum_{t \in S} \bar{f}_t(z_t) - \sum_{t \in S} \bar{f}_t(o) \right] = \mathbb{E}[\chi_m] \cdot \mathbb{E} \left[ \sum_{t=m}^{T} \bar{f}_t(z_{t-m+1}) - \bar{f}_t(o) \right] \leq \mathbb{E}[\chi_m] \cdot R_{Algorithm \ 3}\left( \frac{T}{m} \right).$$

Therefore, we have that

$$\mathbb{E} \left[ \sum_{t=m}^{T} f_t(z_{t-m+1:t}) - \bar{f}_t(o) \right] = \mathbb{E} \left[ \sum_{t=m}^{T} \bar{f}_t(z_{t-m+1}) - \bar{f}_t(o) \right] + \mathbb{E} \left[ \sum_{t=m}^{T} f_t(z_{t-m+1:t}) - \bar{f}_t(z_{t-m+1}) \right]$$

$$\leq (\mathbb{E}[\chi_m])^{-1} R_{Algorithm \ 3}\left( \frac{T}{m} \right) + \mathbb{E} \left[ \sum_{t=m}^{T} f_t(z_{t-m+1:t}) - \bar{f}_t(z_{t-m+1}) \right]$$

$$\leq 3m \cdot R_{Algorithm \ 3}\left( \frac{T}{m} \right) + \mathbb{E} \left[ \sum_{t=m}^{T} f_t(z_{t-m+1:t}) - \bar{f}_t(z_{t-m+1}) \right],$$

where the last inequality follows from

$$\mathbb{E}[\chi_m] = \mathbb{E}[b_m] \prod_{i=1}^{m-1} \mathbb{E}[1 - b_i] = \frac{1}{m} \left( 1 - \frac{1}{m} \right)^{m-1} \geq \frac{1}{em} > \frac{1}{3m}.$$

$\square$

## A.3   Bounding Moving Cost

Lemma 11 and Lemma 12 almost give the regret guarantee in Theorem 6. We are left with bounding the moving cost term in Lemma 12

$$\mathbb{E} \left[ \sum_{t=m}^{T} f_t(z_{t-m+1:t}) - \bar{f}_t(z_{t-m+1}) \right].$$

(Cassel and Koren, 2020) bounded each of the summands by $\mathbb{E}[\delta_t + \nu_t]$, where $\delta_t = \|o_{t+1} - o_t\|_2$, and $\nu_t = \|z_t - o_t\|_2$. Making use of the $\kappa_0$-convexity induced by the affine memory structure of non-stochastic control problems, we can establish a tighter bound that is necessary to obtain optimal regret in Theorem 6.

**Lemma 13** (Moving cost). *Let $(\mathcal{K}, \eta, m, T)$ be the input for Algorithm 1. Suppose that the loss functions $\{f_t\}_{t=m}^{T}$ and the convex compact set $\mathcal{K}$ satisfy Assumption 5, Assumption 6, Assumption 7, and Assumption 8. Suppose $m \leq 2/(\eta\alpha R_H)$. Then, the iterates output by Algorithm 1 satisfy*

$$\mathbb{E}\left[\sum_{t=m}^{T} f_t(z_{t-m+1:t}) - \bar{f}_t(z_{t-m+1})\right] \leq \frac{12m^4\beta R_H d \cdot \max\{2, \eta\alpha R_H\sqrt{T}\}}{\eta\alpha\kappa(G)} \log(\eta\alpha R_H + 1) + \frac{10m^4\beta R_H^3 d\sqrt{T}}{\kappa(G)}$$
$$+ m^2\beta d R_H^3 \sqrt{T} + \eta d G D^2 \beta T.$$

*In particular, with $\eta = \Theta(\sqrt{T})$, $m = \mathrm{poly}(\log T)$ and $\kappa(G) = \Omega(1)$ by Assumption 5, we have that the above bound is of order $\tilde{O}(\sqrt{T})$.*

*Proof of Lemma 13.* We can decompose the moving cost into three terms:

$$\underbrace{\mathbb{E}\left[\sum_{t=m}^{T} f_t(z_{t-m+1:t}) - f_t(o_{t-m+1:t})\right]}_{(a)} + \underbrace{\mathbb{E}\left[\sum_{t=m}^{T} f_t(o_{t-m+1:t}) - \bar{f}_t(o_{t-m+1})\right]}_{(b)} + \underbrace{\mathbb{E}\left[\sum_{t=m}^{T} \bar{f}_t(o_{t-m+1}) - \bar{f}_t(z_{t-m+1})\right]}_{(c)},$$

and we will bound each of the terms separately. In particular, $(a), (c)$ can be seen as perturbation loss suffered by the algorithm during exploration. $(b)$ is the moving cost determined by the stability of the algorithm's neighboring iterates. We start with establishing bounds on $(a), (c)$.

**Perturbation loss.** As before, we denote $\mathcal{F}_t = \sigma(\{b_s, v_s\}_{s=1}^{t})$ to be the $\sigma$-algebra generated by the randomness of Algorithm 1 through sampling $b_s, v_s$ up to time $t$. First, $(c) \leq 0$ by Jensen's inequality for conditional expectations. In particular, recall that $\chi_t = b_t \Pi_{i=1}^{m-1}(1 - b_{t-i})$ denotes whether the decision is updated at time $t$. For every $t \in \mathbb{N}$, denote as $T(t) := \max\{s < t \mid \chi_s = 1\}$ the last time that the algorithm updates its decision. It naturally holds that $o_t = o_{T(t)+1}$, $v_t = v_{T(t)+1}$ by design of Algorithm 1. Therefore, we have

$$\mathbb{E}[\bar{f}_t(o_{t-m+1}) - \bar{f}_t(z_{t-m+1})] = \mathbb{E}[\bar{f}_t(o_{T(t-m+1)+1}) - \bar{f}_t(z_{T(t-m+1)+1})]$$
$$= \mathbb{E}[\bar{f}_t(o_{T(t-m+1)+1}) - \bar{f}_t(o_{T(t-m+1)} + \hat{A}_{t-m}^{-\frac{1}{2}} v_{T(t-m+1)+1})].$$

By the sampling rule in Line 13 of Algorithm 1, if the algorithm updates its decision at time $t$ ($\chi_t = 1$), then $v_{t+1}$ is independent drawn from previous steps. On the other hand, $o_{t+1}$ is measurable w.r.t. $\mathcal{F}_t$. Therefore, we have by Jensen's inequality for conditional expectations that

$$\mathbb{E}[\bar{f}_t(o_{T(t-m+1)+1}) - \bar{f}_t(o_{T(t-m+1)+1} + \hat{A}_{t-m}^{-\frac{1}{2}} v_{T(t-m+1)+1})]$$
$$= \mathbb{E}[\bar{f}_t(o_{T(t-m+1)+1}) - \mathbb{E}[\bar{f}_t(o_{T(t-m+1)+1} + \hat{A}_{t-m}^{-\frac{1}{2}} v_{T(t-m+1)+1}) \mid \mathcal{F}_{T(t-m+1)}]]$$
$$\leq \mathbb{E}[\bar{f}_t(o_{T(t-m+1)+1}) - \bar{f}_t(\mathbb{E}[o_{T(t-m+1)+1} + \hat{A}_{t-m}^{-\frac{1}{2}} v_{T(t-m+1)+1} \mid \mathcal{F}_{T(t-m+1)}])]$$
$$= \mathbb{E}[\bar{f}_t(o_{T(t-m+1)}) - \bar{f}_t(o_{T(t-m+1)})]$$
$$= 0.$$

Summing up over $t$, we get that $(c) \leq 0$. We move on to bound $(a)$. We start with a (conditional) independence argument.

**Independence argument.** Fix $t$. Denote $t_1 = T(t)$ to be the most recent time when the algorithm makes an update. Additionally, denote $t_2 = T(T(t))$ to be the second most recent time when the algorithm makes an update. We already have $o_t = o_{t_1+1}$. By the delayed update rule in Line 12 of Algorithm 1, we have that $o_{t_1+1}$ is updated with $o_{t_1} = o_{t_2+1}$ and gradient information $\tilde{g}_{t_2}$ and thus is $\mathcal{F}_{t_2}$-measurable. On the other hand, consider the sequence of random vectors $v_{t-m+1}, \ldots, v_t$. We have that $v_s = v_{T(s)+1}$ for every $t - m + 1 \leq s \leq t$. Since the algorithm updates at most once every $m$ steps, we have that $T(s) \geq t_2$ for all $t - m + 1 \leq s \leq t$. Therefore, $v_{t-m+1}, \ldots, v_t$ is independent of $\mathcal{F}_{t_2}$. This observation de-correlates $o_t$ with $v_{t-m+1}, \ldots, v_t$ conditioning on $\mathcal{F}_{t_2}$.

For notation simplicity, for matrices $A_1, \ldots, A_n \in \mathbb{R}^{d \times d}$ and vectors $v_1, \ldots, v_n \in \mathbb{R}^d$, we slightly abuse notation and denote as $A_{1:n} v_{1:n} = (A_1 v_1, \ldots, A_n v_n) \in \mathbb{R}^{nd}$ to be the concatenated matrix-vector product of the two sequences.

We apply Taylor's theorem to the function $f_t$ and obtain that

$$(a) = \sum_{t=m}^{T} \mathbb{E}\left[ \nabla f_t(o_{t-m+1:t})^\top \hat{A}_{t-m:t-1}^{-\frac{1}{2}} v_{t-m+1:t} + \frac{1}{2} v_{t-m+1:t}^\top \hat{A}_{t-m:t-1}^{-\frac{1}{2}} \nabla^2 f_t(q_{t-m+1:t}) \hat{A}_{t-m:t-1}^{-\frac{1}{2}} v_{t-m+1:t} \right],$$

where $q_{t-m+1:t}$ is some point that lies on the line segment connecting $o_{t-m+1:t}$ and $z_{t-m+1:t}$. By the conditional independence argument, we have that the first order term vanishes: let $t_2 = T(T(t))$,

$$\mathbb{E}\left[ \nabla f_t(o_{t-m+1:t})^\top \hat{A}_{t-m:t-1}^{-\frac{1}{2}} v_{t-m+1:t} \right] = \mathbb{E}\left[ \mathbb{E}\left[ \nabla f_t(o_{t-m+1:t})^\top \hat{A}_{t-m:t-1}^{-\frac{1}{2}} v_{t-m+1:t} \mid \mathcal{F}_{t_2} \right] \right]$$

$$= \mathbb{E}\left[ \nabla f_t(o_{t-m+1:t})^\top \hat{A}_{t-m:t-1}^{-\frac{1}{2}} \mathbb{E}\left[ v_{t-m+1:t} \mid \mathcal{F}_{t_2} \right] \right]$$

$$= 0.$$

The above sum thus reduces to the sum of second-order terms:

$$\frac{1}{2} \sum_{t=m}^{T} \mathbb{E}\left[ v_{t-m+1:t}^\top \hat{A}_{t-m:t-1}^{-\frac{1}{2}} \nabla^2 f_t(q_{t-m+1:t}) \hat{A}_{t-m:t-1}^{-\frac{1}{2}} v_{t-m+1:t} \right]$$

$$\leq \frac{1}{2} \sum_{t=m}^{T} \mathbb{E}\left[ (\hat{A}_{t-m}^{-\frac{1}{2}} v_{t-m+1}, \ldots, \hat{A}_{t-m}^{-\frac{1}{2}} v_t)^\top \nabla^2 f_t(q_{t-m+1:t}) (\hat{A}_{t-m}^{-\frac{1}{2}} v_{t-m+1}, \ldots, \hat{A}_{t-m}^{-\frac{1}{2}} v_t) \right],$$

$$\leq \frac{1}{2} \sum_{t=m}^{T} \mathbb{E}\left[ \max_{u \in \mathbb{R}^{dm}: \|u\|_2^2 = m} u^\top \begin{bmatrix} \hat{A}_{t-m}^{-\frac{1}{2}} & 0 & \ldots & 0 \\ 0 & \hat{A}_{t-m}^{-\frac{1}{2}} & \ldots & 0 \\ 0 & 0 & \ldots & \hat{A}_{t-m}^{-\frac{1}{2}} \end{bmatrix} \nabla^2 f_t(q_{t-m+1:t}) \begin{bmatrix} \hat{A}_{t-m}^{-\frac{1}{2}} & 0 & \ldots & 0 \\ 0 & \hat{A}_{t-m}^{-\frac{1}{2}} & \ldots & 0 \\ 0 & 0 & \ldots & \hat{A}_{t-m}^{-\frac{1}{2}} \end{bmatrix} u \right],$$

$$\leq \frac{m^2}{2} \sum_{t=m}^{T} \mathbb{E}\left[ \nabla^2 f_t(q_{t-m+1:t}) \cdot \begin{bmatrix} \hat{A}_{t-m}^{-1} & 0 & \ldots & 0 \\ 0 & \hat{A}_{t-m}^{-1} & \ldots & 0 \\ 0 & 0 & \ldots & \hat{A}_{t-m}^{-1} \end{bmatrix} \right],$$

$$= \frac{m^2}{2} \sum_{t=m}^{T} \mathbb{E}\left[ \hat{A}_{t-m}^{-1} \cdot \sum_{i=1}^{m} [\nabla^2 f_t(q_{t-m+1:t})]_{ii} \right],$$

where the first inequality follows from $\hat{A}_s \preceq \hat{A}_t$ for all $s < t$; the second inequality follows from taking maximum over all concatenated unit vectors in $\mathbb{R}^d$; the third inequality follows from the inequality between trace and spectral norm.

From this point on, the bound follows almost identically to the proof of Proposition 21 in (Suggala et al., 2024). We will reiterate the proof in a concise manner for completeness.

Note that by the assumption of affine memory structure (Assumption 5), we can write the following expression for the Hessian matrix of $f_t$ evaluated at $q_{t-m+1:t}$ as the following:

$$\nabla^2 f_t(q_{t-m+1:t}) = \begin{bmatrix} W_{t-m+1}^\top \nabla^2 \ell_t(q) W_{t-m+1} & \ldots & W_{t-m+1}^\top \nabla^2 \ell_t(q) W_t \\ \ldots & \ldots & \ldots \\ W_t^\top \nabla^2 \ell_t(q) W_{t-m+1} & \ldots & W_t^\top \nabla^2 \ell_t(q) W_t \end{bmatrix},$$

where $q = B_t + \sum_{i=0}^{m-1} G^{[i]} Y_{t-i} q_{t-i}$, and $W_{t-m+i} = G^{[m-i]} Y_{t-m+i}, \forall 1 \leq i \leq m$. By the curvature assumption on $\ell_t$ (Assumption 6), we can further bound the sum of diagonal blocks by

$$\sum_{i=1}^{m} [\nabla^2 f_t(q_{t-m+1:t})]_{ii} \preceq \beta \sum_{i=1}^{m} W_{t-m+i}^\top W_{t-m+i}$$

$$= \beta \sum_{i=1}^{m} Y_{t-m+i}^\top (G^{[m-i]})^\top G^{[m-i]} Y_{t-m+i}$$

$$\preceq \beta R_H \sum_{i=1}^{m} Y_{t-m+i}^\top Y_{t-m+i}.$$

Therefore, we can further bound

$$\frac{m^2}{2} \sum_{t=m}^{T} \mathbb{E}\left[\hat{A}_{t-m}^{-1} \cdot \sum_{i=1}^{m} [\nabla^2 f_t(q_{t-m+1:t})]_{ii}\right] \le \frac{m^2 \beta R_H}{2} \sum_{t=m}^{T} \mathbb{E}\left[\hat{A}_{t-m}^{-1} \cdot \left(\sum_{s=t-m+1}^{t} Y_s^\top Y_s\right)\right]$$

$$\le m^2 \beta R_H \sum_{t=m}^{T} \mathbb{E}\left[\hat{A}_t^{-1} \cdot \left(\sum_{s=t-m+1}^{t} Y_s^\top Y_s\right)\right].$$

Consider $\gamma = \lfloor \sqrt{T} \rfloor$ and endpoints $k_j = \gamma(j-1) + m$ for $j = 1, \ldots, J$, and $J = \lfloor \frac{T-m}{\gamma} \rfloor$. Using the fact that $\text{Trace}(AC) \le \text{Trace}(BC)$ for any PSD matrices $A, B, C$ with $A \preceq B$, we can further decompose and bound the sum in the above expression by

$$m^2 \beta R_H \sum_{t=m}^{T} \hat{A}_t^{-1} \cdot \left(\sum_{s=t-m+1}^{t} Y_s^\top Y_s\right) \le m^2 \beta R_H \sum_{j=1}^{J} \hat{A}_{k_j}^{-1} \cdot \left(\sum_{t=k_j}^{k_{j+1}-1} \sum_{s=t-m+1}^{t} Y_s^\top Y_s\right) + m^2 \beta \gamma d R_H^3$$

$$\le m^3 \beta R_H \sum_{j=1}^{J} \hat{A}_{k_j}^{-1} \cdot \left(\sum_{t=k_j-m+1}^{k_{j+1}-1} Y_t^\top Y_t\right) + m^2 \beta \gamma d R_H^3$$

$$\le \frac{2m^3 \beta R_H}{\kappa(G)} \sum_{j=1}^{J} \hat{A}_{k_j}^{-1} \cdot \left(5m R_H^2 I + \sum_{t=k_j}^{k_{j+1}-1} H_t\right) + m^2 \beta \gamma d R_H^3$$

$$= \frac{2m^3 \beta R_H}{\kappa(G)} \sum_{j=1}^{J} \hat{A}_{k_j}^{-1} \cdot \left(\sum_{t=k_j}^{k_{j+1}-1} H_t\right) + \frac{10 m^4 \beta R_H^3 d \sqrt{T}}{\kappa(G)} + m^2 \beta d R_H^3 \sqrt{T}.$$

where the first inequality follows by applying the radius bounds on $Y_t$ for the last $T - J\gamma$ terms and using the fact that $\hat{A}_s \preceq \hat{A}_t$ for any $s < t$; the last inequality follows by applying Proposition 4.8 in (Simchowitz, 2020). Therefore, it suffices to bound the sum in the first term in expectation. First, recall that $\forall t$, $\chi_t = b_t \Pi_{i=1}^{m-1}(1 - b_i)$ denotes whether Algorithm 1 updates during round $t$, and $S = \{m \le t \le T : \chi_t = 1\}$ denotes the set of all rounds where Algorithm 1 updates. To bound the following, we make use of the facts that (1) $H_t$ is oblivious, and (2) $\chi_t$ is independent of $\hat{A}_k^{-1}$ for any $k \le t - m$. Then, we have that

$$\mathbb{E}\left[\sum_{j=1}^{J} \sum_{t \in [k_j, k_{j+1}-1] \cap S} \hat{A}_{k_j-m}^{-1} \cdot H_t\right] = \mathbb{E}\left[\sum_{j=1}^{J} \sum_{t=k_j}^{k_{j+1}-1} \hat{A}_{k_j-m}^{-1} \cdot H_t \cdot \chi_t\right]$$

$$= \sum_{j=1}^{J} \sum_{t=k_j}^{k_{j+1}-1} \mathbb{E}[\hat{A}_{k_j-m}^{-1} \cdot H_t] \cdot \mathbb{E}[\chi_t]$$

$$= \mathbb{E}[\chi_m] \cdot \mathbb{E}\left[\sum_{j=1}^{J} \sum_{t=k_j}^{k_{j+1}-1} \hat{A}_{k_j-m}^{-1} \cdot H_t\right]$$

$$\ge \mathbb{E}[\chi_m] \cdot \mathbb{E}\left[\sum_{j=1}^{J} \sum_{t=k_j}^{k_{j+1}-1} \hat{A}_{k_j}^{-1} \cdot H_t\right].$$

Using this and that $\hat{A}_t \preceq 2\hat{A}_{t-m}$, we have that

$$\mathbb{E}\left[\sum_{j=1}^{J}\sum_{t=k_j}^{k_{j+1}-1}\hat{A}_{k_j}^{-1}\cdot H_t\right] \leq 2\mathbb{E}[\chi_m]^{-1}\mathbb{E}\left[\sum_{j=1}^{J}\sum_{t\in[k_j,k_{j+1}-1]\cap S}\hat{A}_{k_j}^{-1}\cdot H_t\right]$$

$$= \frac{6m}{\eta\alpha}\cdot\mathbb{E}\left[\sum_{j=1}^{J}\hat{A}_{k_j}^{-1}\cdot(\hat{A}_{k_{j+1}-1}-\hat{A}_{k_j-1})\right]$$

$$\leq \frac{6m\cdot\max\{2,\eta\alpha R_H\gamma\}}{\eta\alpha}\cdot\mathbb{E}\left[\sum_{j=1}^{J}\hat{A}_{k_{j+1}-1}^{-1}\cdot(\hat{A}_{k_{j+1}-1}-\hat{A}_{k_j-1})\right]$$

$$\leq \frac{6m\cdot\max\{2,\eta\alpha R_H\gamma\}}{\eta\alpha}\cdot\mathbb{E}\left[\sum_{j=1}^{J}\log\left(\frac{\det(\hat{A}_{k_{j+1}-1})}{\det(\hat{A}_{k_j-1})}\right)\right]$$

$$\leq \frac{6m\cdot\max\{2,\eta\alpha R_H\gamma\}}{\eta\alpha}\cdot\mathbb{E}[\log\det(\hat{A}_T)]$$

$$\leq \frac{6dm\cdot\max\{2,\eta\alpha R_H\gamma\}}{\eta\alpha}\log(\eta\alpha R_H+1),$$

where the first inequality follows from $\hat{A}_{k_{j+1}-1}\preceq\max\{2,\eta\alpha R_H\gamma\}\hat{A}_{k_j}$, and rest follows from the standard inequalities used in Newton-step analysis (Hazan et al., 2007).

Combining all the bounds, we have that

$$(a) \leq \frac{12m^4\beta R_H d\cdot\max\{2,\eta\alpha R_H\sqrt{T}\}}{\eta\alpha\kappa(G)}\log(\eta\alpha R_H+1) + \frac{10m^4\beta R_H^3 d\sqrt{T}}{\kappa(G)} + m^2\beta dR_H^3\sqrt{T}.$$

**Movement cost.** By design, the algorithm updates at most once in every $m$ iterations. Therefore,

$$\|(o_{t-m+1},\ldots,o_t)-(o_{t-m+1},\ldots,o_{t-m+1})\|_2 \leq s,$$

where $s$ is the Euclidean distance between neighboring iterates in Algorithm 3. By analysis in Lemma 11, we have $s \leq \eta dGD$. By Lipschitz assumption on $f_t$, we have

$$(b) \leq \eta dGD^2\beta T.$$

Combining, we have that the total moving cost is bounded by

$$\mathbb{E}\left[\sum_{t=m}^{T}f_t(z_{t-m+1:t})-\bar{f}_t(z_{t-m+1})\right] \leq \frac{12m^4\beta R_H d\cdot\max\{2,\eta\alpha R_H\sqrt{T}\}}{\eta\alpha\kappa(G)}\log(\eta\alpha R_H+1) + \frac{10m^4\beta R_H^3 d\sqrt{T}}{\kappa(G)}$$

$$+ m^2\beta dR_H^3\sqrt{T} + \eta dGD^2\beta T.$$

$\square$

### A.4    Proof of Theorem 6

Combining result from Lemma 11, Lemma 12, and Lemma 13, we have the regret of Algorithm 1 w.r.t. to any $z \in \mathcal{K}$ is bounded by

$$\mathbb{E}[\text{Regret}_T(z)] \leq 3m\left(\frac{2\beta d}{\eta\alpha}\log(\eta R_H T+1)+2d_0(GD)+\frac{D^2 d_0 R_H}{2\eta}+3\eta d_0 d^2(GD)^2 R_H T\right)$$

$$+ \frac{12m^4\beta R_H d\cdot\max\{2,\eta\alpha R_H\sqrt{T}\}}{\eta\alpha\kappa(G)}\log(\eta R_H T+1) + \frac{10m^4\beta R_H^3 d\sqrt{T}}{\kappa(G)}$$

$$+ m^2\beta dR_H^3\sqrt{T} + \eta dG\beta D^2 T,$$

and by choosing $\eta = \Theta\left(\frac{1}{\alpha\sqrt{T}}\right)$, we have the regret above is bounded by $\tilde{O}\left(\frac{\beta}{\alpha}GD\sqrt{T}\right)$.

# B  Proof of Lemma 9

In this section, we prove Lemma 9. We begin by defining the Markov operator (Simchowitz et al., 2020).

**Definition 14** (Markov operator). Given a partially observable LDS instance parametrized by dynamics $A \in \mathbb{R}^{d_x \times d_x}, B \in \mathbb{R}^{d_x \times d_u}, C \in \mathbb{R}^{d_y \times d_x}$ satisfying Assumption 2 with $(\kappa, \gamma)$-strongly stabilizing $K$, define the Markov operator to be a sequence of matrices $G = \{G^{[i]}\}_{i \geq 0}$ such that $G^0 = [0; I_{d_u}]$ and $\forall i > 0, G^{[i]}$ is given by

$$G^{[i]} = \begin{bmatrix} C \\ KC \end{bmatrix} (A + BKC)^{i-1} B \in \mathbb{R}^{(d_y + d_u) \times d_u}.$$

The next observation (Observation 15) relates $y_t(K)$, the signals used by the DRC policies (Definition 1), to the observations and controls along the learner's trajectory. This justifies the accessibility of the signals.

**Observation 15.** *Given a partially observable LDS instance with $(\kappa, \gamma)$-strongly stabilizing $K$, let $G$ be the Markov operator in Definition 14. For $t, m \in \mathbb{N}$, let $M_1, \ldots, M_{t-1} \in \mathbb{R}^{m \times d_u \times d_y}$ be $(t-1)$ DRC matrices (Definition 1). Let $(y_t, u_t)$ be the observation-control pair reached by playing $M_1, \ldots, M_{t-1}$ for time step $t = 1, \ldots, t-1$. Let $u_t(K)$ be the control at time $t$ by executing the linear policy $K$, i.e. $y_t(K) = Ku_t(K)$, and $y_t(K)$ be the would-be observation had $K$ been executed from the beginning of the time. Then, $(y_t, u_t)$ and $(y_t(K), u_t(K))$ can be related by the following equality:*

$$\begin{bmatrix} y_t \\ u_t \end{bmatrix} = \begin{bmatrix} y_t(K) \\ Ky_t(K) \end{bmatrix} + \sum_{i=1}^{t} G^{[i]} \left( \sum_{j=0}^{m-1} M_{t-i}^{[j]} y_{t-i-j}(K) \right). \tag{9}$$

*In particular, Eq. (9) implies that $y_t(K)$ can be computed by the learner through access to the Markov operator $G$ and the observations along the learner's own trajectory.*

With Eq. (9), we are almost ready to reduce the control instance to the BCO-M problem. One delicate detail is that the BCO-M problem is defined for vector-valued decisions, and the the space $\mathcal{M}(m, R_{\mathcal{M}})$ is a space of sequences of matrices. For clarity of presentation, we define the following embedding operators.

**Definition 16** (Embedding operators). For $m, d_y, d_u \in \mathbb{N}$, denote $d = md_y d_u$. The embedding operator $\mathfrak{e} : (\mathbb{R}^{(d_u \times d_y)})^m \to \mathbb{R}^d$ is the natural embedding of a DRC controller $M = M^{[0:m-1]}$ (Definition 1) in $\mathbb{R}^d$. In particular, $\forall k \in [m-1], i \in [d_u], j \in [d_y]$,

$$\langle e_{kd_u d_y + (i-1)d_y + j}, \mathfrak{e}(M) \rangle = M_{ij}^{[k]}.$$

Let $\mathfrak{e}_y : (\mathbb{R}^{d_y})^m \to \mathbb{R}^{d_u \times d}$ be given by $\forall y_{t-m+1}, \ldots, y_t \in \mathbb{R}^{d_y}, \forall i \in [d_u], k \in [m]$,

$$[\mathfrak{e}_y(y_{t-m+1:t})]_{i, j+1:j+d_y} = y_{t-k+1}, \quad \text{if} \quad j = (k-1)d_u d_y + (i-1)d_y.$$

*Proof of Lemma 9.* Let $B_t = (y_t(K), Ky_t(K)) + \sum_{i=m}^{t} G^{[i]} Y_{t-i} \mathfrak{e}(M_{t-i}) \in \mathbb{R}^{d_u + d_y}, \forall t$. Let $Y_t = \mathfrak{e}_y(y_{t-m+1:t}(K))$. Then, by Eq. (9), we have

$$c_t(y_t, u_t) = c_t \left( B_t + \sum_{i=0}^{m-1} G^{[i]} Y_{t-i} \mathfrak{e}(M_{t-i}) \right).$$

We can't directly define our $f_t$ as this function because the $\sum_{i=m}^{t} G^{[i]} Y_{t-i} \mathfrak{e}(M_{t-i})$ term in $B_t$ depends on historical steps, which will lead to an unbounded memory.

To this end, we define

$$c_t \left( (y_t(K), Ky_t(K)) + \sum_{i=0}^{m-1} G^{[i]} Y_{t-i} \mathfrak{e}(M_{t-i}) \right) =: f_t(\mathfrak{e}(M_{t-m+1}), \ldots, \mathfrak{e}(M_t)), \tag{10}$$

which is independent of $M_{t-m+1:t}$, and $Y_t$ is independent of $M_{1:T}$. Note that by the loss function with memory has an affine memory structure. Moreover, $G^{[i]}$ and $Y_{t-i}$ are computable by the learner given system parameters. By the choice of $m = \Theta(\log T)$, the Lipschitzness of $c_t$, and the norm-decaying property of $G^{[i]}$ due to the stability of the system, we can bound the distance

$$|c_t(y_t, u_t) - f_t(\mathfrak{c}(M_{t-m+1}), \ldots, \mathfrak{c}(M_t))| = O\left(\frac{1}{\text{poly}(T)}\right).$$

In particular, we can choose $m$ such that this term is $O\left(\frac{1}{T^2}\right)$, then any regret bound on $f_t$ in the BCO-M setting directly transfers to the same regret bound on $c_t$ in the control setting, at a negligible $o(1)$ cost which will be subsumed by the approximation error term.

We are left with two steps to conclude this lemma. First, we need to show that $f_t$ indeed satisfies the conditions in Definition 5 and specify the constants. Next, because $f_t$ is not equal to $c_t$ while we only have access to $c_t$, the gradient estimator for $f_t$ constructed from $c_t$ is biased. We need to bound this error and show that it has negligible impact on the regret bound.

**Step 1:** We denote the unary form to be $\bar{f}_t$ as in Definition 2 . Let $\mathcal{O} = \{\mathcal{M}(m, R_{\mathcal{M}}), m, \{f_t\}_{t \geq m, t \in \mathbb{N}}\}$ be the associated BCO-M instance (Definition 5).

By construction, $f_t$ satisfies Assumption 8 and Assumption 5 other than the assumption on positive convolution invertibility-modulus if the truncated vector $(y_t(K), Ky_t(K)) + \sum_{i=0}^{m-1} G^{[i]} Y_{t-i} \mathfrak{c}(M_{t-i})$ always lives in the $R\mathbb{B}_{d_y+d_u}$ and the matrix parameters are bounded. For the positive convolution invertibility-modulus, Lemma 3.1 in (Simchowitz, 2020) proves that the $G^{[0]}, \ldots, G^{[m-1]}$ induced by the Markov operator in Definition 14 satisfies the positive convolution invertibility-modulus, i.e. $\kappa(G) = \Omega(1)$. By assumption on $c_t$, the curvature assumption in Assumption 6 is also satisfied. It is left to check the diameter bounds in Assumption 5 and Assumption 7.

First, we establish a diameter bound on our learning comparator set $\mathcal{K} = \mathcal{M}(m, R_{\mathcal{M}})$.

**Diameter bound on $\mathcal{K} = \mathcal{M}(m, R_{\mathcal{M}})$.** The diameter of the set $\mathcal{M}(m, R_{\mathcal{M}})$ is given by

$$\max_{M_1, M_2 \in \mathcal{M}(m, R_{\mathcal{M}})} \|\mathfrak{c}(M_1) - \mathfrak{c}(M_2)\|_2 \leq \sqrt{m} \max_{M_1, M_2 \in \mathcal{M}(m, R_{\mathcal{M}})} \max_{0 \leq j \leq m-1} \|M_1^{[j]} - M_2^{[j]}\|_F$$

$$\leq \sqrt{m \max\{d_u, d_y\}} \max_{M_1, M_2 \in \mathcal{M}(m, R_{\mathcal{M}})} \max_{0 \leq j \leq m-1} \|M_1^{[j]} - M_2^{[j]}\|_{\text{op}}$$

$$\leq \sqrt{m \max\{d_u, d_y\}} R_{\mathcal{M}}.$$

Algorithm 2 essentially calls the improper BCO-M algorithm in Algorithm 1. With the diameter bound on $\mathcal{M}(m, R_{\mathcal{M}})$, we know that the matrices governing the controls picked by Algorithm 2 during each round lives in the set $\mathcal{M}(m, R_{\mathcal{M}} + m)$. For simplicity, we may assume that $R_{\mathcal{M}} \geq m$ and thus the decisions made by Algorithm 2 live in $\mathcal{M}(m, 2R_{\mathcal{M}})$.

Now, we are ready to prove the gradient bound on $f_t$ and the bounds on the observation-control pairs produced by Algorithm 2.

**Gradient bound of $f_t$.** First, we bound the sum of operator norms for the Markov operator. By Assumption 2 and the fact that $\forall M \in \mathbb{R}^{n \times n}, \|M\|_{\text{op}} \leq \sqrt{n}\|M\|_{\text{max}} \leq \sqrt{n}\|M\|_2$, we have that

$$\sum_{i=0}^{\infty} \|G^{[i]}\|_{\text{op}} \leq 1 + \sum_{i=1}^{\infty} \sqrt{\|C(A + BKC)^{i-1}B\|_{\text{op}}^2 + \|KC(A + BKC)^{i-1}B\|_{\text{op}}^2}$$

$$\leq 1 + \sqrt{d_x(1 + \kappa^2)}\kappa_{\text{sys}}^2 \sum_{i=1}^{\infty} \|HL^{i-1}H^{-1}\|_2$$

$$\leq 1 + \sqrt{d_x(1 + \kappa^2)}\kappa_{\text{sys}}^2 \sum_{i=0}^{\infty} (1 - \gamma)^i$$

$$= 1 + \frac{\sqrt{d_x(1 + \kappa^2)}\kappa_{\text{sys}}^2}{\gamma}.$$

The signals $y_t(K)$ has the following norm bound:

$$\max_{t\in[T]}\|y_t(K)\|_2 = \max_{t\in[T]}\left\|C\sum_{i=0}^{t-1}(A+BKC)^i w_{t-i} + e_t\right\|_2$$

$$\leq R_{w,e}\left(1 + \sqrt{d_x}\kappa_{\mathbf{sys}}\sum_{i=1}^{\infty}\|HL^{i-1}H^{-1}\|_2\right)$$

$$\leq R_{w,e}\left(1 + \frac{\sqrt{d_x}\kappa_{\mathbf{sys}}}{\gamma}\right).$$

By the unfolding of $(y_t, u_t)$ in Eq. (9), the observation and control pair $(y_t, u_t)$ has the following norm bound: $\forall t$,

$$\|(y_t, u_t)\|_2 \leq \|(y_t(K), Ky_t(K))\|_2 + \left\|\sum_{i=1}^{t}G^{[i]}\left(\sum_{j=0}^{m-1}M_{t-i}^{[j]}y_{t-i-j}(K)\right)\right\|_2$$

$$\leq R_{w,e}\left(1 + \frac{\sqrt{d_x}\kappa_{\mathbf{sys}}}{\gamma}\right)\left(\sqrt{1+\kappa^2} + 2R_{\mathcal{M}}\left\|\sum_{i=0}^{\infty}G^{[i]}\right\|_{\mathrm{op}}\right)$$

$$\leq R_{w,e}\left(1 + \frac{\sqrt{d_x}\kappa_{\mathbf{sys}}}{\gamma}\right)\left(\sqrt{1+\kappa^2} + 2R_{\mathcal{M}}\left(1 + \frac{\sqrt{d_x(1+\kappa^2)}\kappa_{\mathbf{sys}}^2}{\gamma}\right)\right) =: R.$$

The above bound holds similarly for the truncated $(\hat{y}_t, \hat{u}_t) := (y_t(K), Ky_t(K)) + \sum_{i=0}^{m-1}G^{[i]}Y_{t-i}\mathfrak{c}(M_{t-i})$ in Eq. (10), which shows that $(\hat{y}_t, \hat{u}_t) \in R\mathbb{B}_{d_y+d_u}$. This also completes checking Assumption 5.

By assumption, $c_t$ obeys the Lipschitz bounds

$$\|\nabla c_t(y,u)\|_2 \leq G_c\|(y,u)\|_2.$$

The gradient bound on $f_t$ is given by

$$\|\nabla f_t(\mathfrak{c}(M_{t-m+1}),\ldots,\mathfrak{c}(M_t))\|_2$$

$$= \left\|((G^{[m-1]}Y_{t-m+1})^\top\nabla c_t(\hat{y}_t,\hat{u}_t),\ldots,(G^{[0]}Y_t)^\top\nabla c_t(\hat{y}_t,\hat{u}_t)\right\|_2$$

$$\leq \sqrt{m}\left(1 + \frac{\sqrt{d_x(1+\kappa^2)}\kappa_{\mathbf{sys}}^2}{\gamma}\right)G_c R_{w,e}^2\left(1 + \frac{\sqrt{d_x}\kappa_{\mathbf{sys}}}{\gamma}\right)^2\left(\sqrt{1+\kappa^2} + 2R_{\mathcal{M}}\left(1 + \frac{\sqrt{d_x(1+\kappa^2)}\kappa_{\mathbf{sys}}^2}{\gamma}\right)\right)^2$$

$$\leq \frac{4096\sqrt{m}G_c R_{w,e}^2 R_{\mathcal{M}}^2 d_x^{2.5}\kappa^3\kappa_{\mathbf{sys}}^8}{\gamma^5}.$$

**Step 2:** Clearly the gradient estimator obtained for $f_t$ is biased. We denote this error as $a_t$ which has a norm bound $\|a_t\|_2 = O(\frac{1}{T})$ with the same reasoning on bounding $|c_t - f_t|$. Now, we only need to show that Algorithm 1 can tolerate such perturbation with no loss in regret.

It's known that the OGD algorithm can tolerate per-round adaptive gradient perturbation with norm bounded by $O(\frac{1}{T})$, with no cost in regret. We will prove a similar argument for Algorithm 3. Suppose in line 6, the unbiased gradient estimator $\tilde{g}_t$ is replaced by a biased estimator $\hat{g}_t = \tilde{g}_t + a_t$, then we want to extend the regret bound for Algorithm 3 in Lemma 11 in this setting with an additional term depending on the sum of magnitude of $a_t$. The analysis in the proof of Lemma 11 remains unchanged except when we bound the regret for the sequence $\{o_t\}_{t=1}^T$ in Eq. (6), the inequality becomes

$$\mathbb{E}\left[\sum_{t=1}^T \bar{f}_t(o_t) - \bar{f}_t(o)\right] \leq \sum_{t=1}^T\mathbb{E}[\nabla\bar{f}_t(o_t)^\top(o_t - o)] - \frac{\alpha_f}{2}\sum_{t=1}^T\mathbb{E}[\|o_t - o\|_{H_t}^2]$$

$$= \sum_{t=1}^T\mathbb{E}[\hat{g}_t^\top(o_t - o)] - \sum_{t=1}^T\mathbb{E}[a_t^\top(o_t - o)] - \frac{\alpha_f}{2}\sum_{t=1}^T\mathbb{E}[\|o_t - o\|_{H_t}^2].$$

The rest of proof after Eq. (6) is indentical, except that we replace each appearance of $\tilde{g}_t$ by $\hat{g}_t$.

Therefore, the regret incurred by using $\hat{g}_t$ instead of $\tilde{g}_t$ as the gradient estimator is bounded by the regret upper bound in Lemma 11 and the additional term $\sum_{t=1}^{T} \mathbb{E}[a_t^\top (o_t - o)] \leq aDT$, where $a = \max_{t \in [T]} \|a_t\|_2$. Since $a = O(\frac{1}{T})$, the additional regret is bounded by $O(1)$.

**Concluding the lemma:** Denote

$$\alpha_f = \alpha_c, \quad \beta_f = \beta_c, \quad D = \sqrt{m \max\{d_u, d_y\}} R_{\mathcal{M}},$$

$$G_f = \frac{4096\sqrt{m} G_c R_{w,e}^2 R_{\mathcal{M}}^2 d_x^{2.5} \kappa^3 \kappa_{\text{sys}}^8}{\gamma^5}.$$

We have that by Assumption 4 that

$$\mathbb{E}\left[\text{Regret}_T^{\mathcal{A}^{\text{NC}}}(\mathcal{L})\right]$$

$$= \max_{M \in \mathcal{M}(m, R_{\mathcal{M}})} \mathbb{E}\left[\sum_{t=1}^{T} c_t(y_t(M_{1:t-1}), u_t(M_{1:t-1})) - c_t(y_t(M), u_t(M))\right]$$

$$\leq G_f Dm + \mathbb{E}\left[\sum_{t=m}^{T} f_t(\mathfrak{e}(M_{t-m+1}), \ldots, \mathfrak{e}(M_t)) - \bar{f}_t(\mathfrak{e}(M))\right]$$

$$+ \mathbb{E}\left[\sum_{t=m}^{T} \bar{f}_t(\mathfrak{e}(M)) - c_t(y_t(M), u_t(M))\right]$$

$$= \mathbb{E}\left[\sum_{t=m}^{T} c_t\left(B_t + \sum_{i=0}^{m-1} G^{[i]} Y_{t-i}\mathfrak{e}(M))\right) - c_t\left((y_t(K), Ky_t(K)) + \sum_{i=0}^{t} G^{[i]} Y_{t-i}\mathfrak{e}(M))\right)\right]$$

$$+ \mathbb{E}\left[\sum_{t=m}^{T} f_t(\mathfrak{e}(M_{t-m+1}), \ldots, \mathfrak{e}(M_t)) - \bar{f}_t(\mathfrak{e}(M))\right] + G_f Dm,$$

where we have

$$c_t\left(B_t + \sum_{i=0}^{m-1} G^{[i]} Y_{t-i}\mathfrak{e}(M))\right) - c_t\left((y_t(K), Ky_t(K)) + \sum_{i=0}^{t} G^{[i]} Y_{t-i}\mathfrak{e}(M))\right)$$

$$\leq G_c \left\|\sum_{i=m}^{t} G^{[i]} Y_{t-i}\mathfrak{e}(M)\right\|_2 \leq G_c \sum_{i=m}^{t} \|G^{[i]}\|_{\text{op}} \|Y_{t-i}\|_F \|M\|_F.$$

Recall that

$$\|M\|_F^2 = \sum_{j=0}^{m-1} \|M^{[j]}\|_F^2 \leq \max\{d_u, d_y\} \sum_{j=0}^{m-1} \|M^{[j]}\|_{\text{op}}^2 \leq m \max\{d_u, d_y\} R_{\mathcal{M}}^2,$$

$$\max_{t \in [T]} \|Y_t\|_2 = \sqrt{m d_y d_u^2} \cdot \max_{t \in [T]} \|y_t(K)\|_2 \leq \sqrt{m d_y d_u^2} \cdot R_{w,e}\left(1 + \frac{\sqrt{d_x} \kappa_{\text{sys}}}{\gamma}\right),$$

$$\sum_{i=m}^{\infty} \|G^{[i]}\|_{\text{op}} \leq \sqrt{d_x(1 + \kappa^2)} \kappa_{\text{sys}}^2 \sum_{i=m}^{\infty} (1 - \gamma)^i \leq \frac{\sqrt{d_x(1 + \kappa^2)} \kappa_{\text{sys}}^2 (1 - \gamma)^m}{\gamma}.$$

We have for $m = \Theta\left(\log T / \log(1/(1 - \gamma))\right)$,

$$G_c \sum_{i=m}^{t} \|G^{[i]}\|_{\text{op}} \|Y_{t-i}\|_F \|M\|_F \leq \frac{4G_c m d_y d_u^2 d_x (1 + \kappa^2) \kappa_{\text{sys}}^4 R_{\mathcal{M}} (1 - \gamma)^m}{\gamma^2}$$

$$\leq \frac{G_c m d_y d_u^2 d_x \kappa^2 \kappa_{\text{sys}}^4 R_{\mathcal{M}}}{\gamma^2 T}.$$

Combining, we have

$$\mathbb{E}\left[\text{Regret}_T^{\mathcal{A}^{\text{NC}}}(\mathcal{L})\right] \le \mathbb{E}\left[\text{Regret}_T^{\mathcal{A}^{\text{B}}}(\mathcal{O})\right] + \frac{G_c m d_y d_u^2 d_x \kappa^2 \kappa_{\text{sys}}^4 R_{\mathcal{M}}}{\gamma^2} + G_f Dm$$

$$\le \mathbb{E}\left[\text{Regret}_T^{\mathcal{A}^{\text{B}}}(\mathcal{O})\right] + 2G_f Dm,$$

which concludes the claim that $\mathcal{O}$ $2G_f DmT^{-1}$-approximates $\mathcal{L}$. $\quad\square$

