# OpenReview forum: "Tight Rates for Bandit Control Beyond Quadratics"
_NeurIPS.cc/2024/Conference — NeurIPS 2024 poster_

### Official Review · Reviewer_iRqM · 2024-07-08

**Soundness:** 3
**Presentation:** 3
**Contribution:** 2
**Rating:** 6
**Confidence:** 3

**Summary:**

This paper studys online control with adversarial pertubations, bandit feedback and adversarial strongly-convex smooth cost functions. This setting is more general than previous works and the authors successfully achieve $O(\sqrt{T})$ regret by leveraging occasional update and Newton-based update.

**Strengths:**

1. This paper generalizes previous settings and get the optimal regret.

2. This paper is well-written and the intuition behind algorithm is explained clearly.

**Weaknesses:**

The technical contribution seems not strong, the analysis of Algorithm 1 (reduce to no-memory BCO, bound regret for base algorithm and moving cost) follows the proof sketch of [1]. The main change is replacing the base algorithm with Newton-based updates in Algorithm 2 of [2], ensuring a tighter bound by utilizing the $\kappa$-convexity and affine memory.


[1] Cassel, A. and Koren, T. (2020). Bandit linear control. Advances in Neural Information Processing Systems, 33:8872–8882.

[2] Suggala, A., Sun, Y. J., Netrapalli, P., and Hazan, E. (2024). Second order methods for bandit
optimization and control. arXiv preprint arXiv:2402.08929.

**Questions:**

What is the main techinical challenge when combining and adapting the proof in [1] and [2]?

**Limitations:**

Yes.

---

> ### Author Rebuttal · Authors · 2024-08-01
>
> Thank you for your insights and valuable feedback! We will address your concerns here.
>
> **Technical contribution**: we consider the main contribution of this work as pushing the frontier of bandit online control, by achieving optimal regret without quadratic loss or stochastic noise assumptions. To achieve this goal, we introduce several new algorithmic/analysis components towards previous works, but they are not the main objective. One technical challenge we resolve is that the analysis in (Suggala et. al. 2024) depends on the delayed dependence of the iterates (the current iterate only depends on the algorithm's sampling randomness up to time $t-m$). The algorithm in (Suggala et. al. 2024) automatically guarantees such independence since each iterate is updated with information up to the $t-m$ step. The limitation of such algorithm is that the gradient estimator only satisfies low-bias guarantee for quadratic functions, and that motivates the adaptation of the reduction in (Cassel and Koren, 2020). However, we note that the update rules used in (Cassel and Koren, 2020) no longer guarantees the delayed independence. Here, we use a delaying mechanism to preserve this delayed independence.

---

> > ### Comment · Reviewer_iRqM · 2024-08-10
> >
> > I thank the authors for the response. I do not have further questions now and will keep my score.

---

### Official Review · Reviewer_vynp · 2024-07-12

**Soundness:** 3
**Presentation:** 3
**Contribution:** 3
**Rating:** 6
**Confidence:** 3

**Summary:**

This paper considers the problem of online non-stochastic control, focusing specifically on scenarios where the loss function is characterized by bandit feedback, strong convexity, and smoothness, and the noise is adversarial. Prior research has typically managed to achieve $O(\sqrt{T})$ regret under assumptions such as quadratic loss functions, full information feedback, or stochastic noise. This paper breaks these assumptions, demonstrating that an $O(\sqrt{T})$ regret bound can be achieved even in the presence of adversarial noise, bandit feedback, and strongly convex loss functions.

**Strengths:**

This paper presents a direct theoretical improvement, offering significant advancements in the field. It is well-written and theoretically solid, providing a robust analysis and clear insights into the online non-stochastic control problem with adversarial noise, bandit feedback, and strongly convex loss functions.

**Weaknesses:**

1. The citation for Optimal rates for bandit non-stochastic control is incorrect; it was mistakenly written as NeurIPS 2024.

2.This paper could benefit from some additional discussion. While this work presents a significant improvement in a specific scenario of online bandit control, it is equally important to address the challenge of designing a single algorithm that can achieve theoretical guarantees across different scenarios simultaneously. For instance, you might consider the problem proposed by the recent work "Handling Heterogeneous Curvatures in Bandit LQR Control" from ICML 2024. I believe that discussing this issue in the related work and future work sections would add significant value to the paper.

**Questions:**

No questions.

**Limitations:**

See weaknesses.

---

> ### Author Rebuttal · Authors · 2024-08-01
>
> Thank you for your insights and valuable feedback. We will address your concerns here.
>
> **Incorrect citation of Sun et. al. (2023)**: Thank you for pointing this out! We will fix the typo accordingly.
>
> **Discussions of previous work**: Thank you for bringing the relevant paper into our attention! Designing adaptive/universal algorithms is a central research topic in online learning, and it's even more important for online control which can be seen as a practical application of online learning. We will add more comparisons with this work and other related works, and discuss potential future directions on making our algorithm universal.
>
> To our understanding, you have no further concerns beyond the two points raised above. We kindly ask you to consider raising the score if the two weaknesses have been addressed. Thank you again for your valuable time!

---

### Official Review · Reviewer_r1Ah · 2024-07-12

**Soundness:** 3
**Presentation:** 3
**Contribution:** 3
**Rating:** 6
**Confidence:** 3

**Summary:**

This paper studies the Linear Quadratic Control (LQC) problem with adversarial perturbations, bandit feedback models, and non-quadratic cost. The authors propose an algorithm that achieves $\mathcal{O}(\sqrt{T})$ optimal regret for bandit non-stochastic control with strongly-convex and smooth cost functions in the presence of adversarial perturbations, which improves the known $\mathcal{O}(T^{2/3})$ of the previous work of Cassel and Koren [2020].

The dynamic system (partially observable linear time-invariant (LTI)) is defined as in Eq.(1). This work is largely inspired by the previous work of Suggala et al. [2024] which achieves optimal regret guarantee in a more restricted setting.

**Strengths:**

1. Though I have skimmed the proof of several lemmas, the analysis part seems to be rigorous and mathematically correct.
2. The delayed mechanism to de-correlate the recent m iterates looks interesting, which may be of use in the other delayed feedback setting.

**Weaknesses:**

1. The specific contribution of this work towards the previous work of Suggala et al. [2024]  is still a little bit unclear. According to Line 382 to 387, it seem that the most important algorithmic contribution is the delay mechanism.
2. Not certain what it means by "preserves an estimation of Hessian $H_t$ for free" in Line 236. It seems related to Assumption 5 which provides the $H_t$ to the learner directly at the end of each iteration. I wonder if this sort of assumptions is general, and whether it is reasonable in the LTI control problem.

Typo:
1. line 248, length to lengthy.
2. Definition 3, $f_t$ should be $f$?

Other than these two issues, I haven't observed any specific weaknesses in this work.

**Questions:**

The questions are raised in the weakness section. I am willing to re-evaluate the scores if these questions are properly answered.

**Limitations:**

This paper is pure theoretical and does not have any limitations.

---

> ### Author Rebuttal · Authors · 2024-08-01
>
> Thank you for your insights and valuable feedback. We will address your concerns here.
>
> **Contribution towards previous work (Suggala et. al. (2024))**: The algorithm and guarantees presented by Suggala et al. (2024) are limited to quadratic functions due to their reliance on a gradient estimator for with-memory loss functions, which maintains a low-bias guarantee only for quadratic functions. We overcome this restriction by reducing the problem to a no-memory optimization scenario, leveraging techniques from Cassel and Koren (2020). In applying Cassel and Koren's reduction technique, a delay mechanism is essential to de-correlate the iterates.
>
> **Clarification of Line 236**: This sentence means $H$ can be directly computed by the system parameters in our framework, avoiding the typical costly sampling for approximation of Hessians (inefficiency is the main obstacle in using second-order methods). This is proved in Lemma 9, where we showed that the control problem is a well-conditioned instance of BCO-M problem. Here, well-conditioned requires Assumption 5.
>
> In the general setting of BCO-M, Assumption 5 is not common. However, we are interested in solving the bandit linear control problem in which this assumption is natural (BCO-M is merely a tool to solve this control problem). In fact, this condition should be understood as a natural feat of LTI control, and only becomes an "assumption" in BCO-M.
>
> Even when the system is unknown, we can run one of the existing system estimation algorithms to obtain system estimates and compute this matrix using the system estimates. The theoretical guarantees for unknown systems are out of scope in this paper, but such extensions have been seen previously in Simchowitz (2020) and Suggala et. al. (2024), and these works have all assumed similar assumptions as in Assumption 5. We will add more discussion about this point in the main text.
>
> [1] Simchowitz, Max. Making non-stochastic control (almost) as easy as stochastic. Advances in Neural Information Processing Systems 33 (2020): 18318-18329.
>
> [2] Suggala, Arun, et al. Second Order Methods for Bandit Optimization and Control. The Thirty Seventh Annual Conference on Learning Theory. PMLR, 2024.
>
> For the typos, we will fix them accordingly. Thank you for the careful reading of our work!
>
> If our response has addressed your concerns, please consider reevaluate our paper. If you have further questions, please let us know. Thank you again for your valuable time and insights!

---

> > ### Comment · Reviewer_r1Ah · 2024-08-13
> >
> > Thank the authors for their response. My questions are well-addressed ,and I would like to increase my score.

---

### Decision · Program_Chairs · 2024-09-25

**Decision:**

Accept (poster)

**Comment:**

This work provided the first T^{1/2} regret result for bandit non-stochastic control for strongly convex loses, where previous results achieving the same rate only applied in presence of stochastic (or smoothed adversarial) perturbations, or for strongly convex quadratics. This contribution was appreciated by the reviewers. Happy to recommend this for acceptance.